# Deciphering Spatio-Temporal Graph Forecasting: A Causal Lens and Treatment

**Yutong Xia**[1]**, Yuxuan Liang**[2],*** Haomin Wen**[3]**, Xu Liu**[1]**, Kun Wang**[4]**,**
**Zhengyang Zhou**[4]**, Roger Zimmermann**[1]
[1]National University of Singapore
[2]The Hong Kong University of Science and Technology (Guangzhou)
[3]Beijing Jiaotong University [4]University of Science and Technology of China
`{yutong.x,yuxliang}@outlook.com;{liuxu,rogerz}@comp.nus.edu.sg`
`wenhaomin@bjtu.edu.cn;wk520529@mail.ustc.edu.cn;zzy0929@ustc.edu.cn`

## Abstract

Spatio-Temporal Graph (STG) forecasting is a fundamental task in many real-world applications. Spatio-Temporal Graph Neural Networks have emerged as the most popular method for STG forecasting, but they often struggle with temporal out-of-distribution (OoD) issues and dynamic spatial causation. In this paper, we propose a novel framework called CaST to tackle these two challenges via causal treatments. Concretely, leveraging a causal lens, we first build a structural causal model to decipher the data generation process of STGs. To handle the temporal OoD issue, we employ the back-door adjustment by a novel disentanglement block to separate the temporal environments from input data. Moreover, we utilize the front-door adjustment and adopt edge-level convolution to model the ripple effect of causation. Experiments results on three real-world datasets demonstrate the effectiveness of CaST, which consistently outperforms existing methods with good interpretability. Our source code is available at https://github.com/yutong-xia/CaST.

## 1 Introduction

Individuals enter a world with intrinsic structure, where components interact with one another across space and time, leading to a spatio-temporal composition. *Spatio-Temporal Graph* (STG) has been pivotal for incorporating this structural information into the formulation of real-world issues. Within the realm of smart cities [60], STG forecasting (e.g., traffic prediction [58, 49, 17, 52] and air quality forecasting [28, 48, 27]) has become instrumental in informed decision-making and sustainability. With recent advances in deep learning, *Spatio-Temporal Graph Neural Networks* (STGNNs) [50, 20] have become the leading approach for STG forecasting. They primarily use Graph Neural Networks (GNN) [23] to capture spatial correlations among nodes, and adopt Temporal Convolutional Networks (TCN) [2] or Recurrent Neural Networks (RNN) [13] to learn temporal dependencies.

However, STG data is subject to temporal dynamics and may exhibit various data generation distributions over time, also known as *temporal out-of-distribution* (OoD) issues or *temporal distribution shift* [6, 67]. As depicted in Figure 1a, the training data (periods A and B) and test data derive from different distributions, namely $P_A(x) \neq P_B(x) \neq P_{test}(x)$. Most prior studies [26, 61, 56, 15, 30] have overlooked this essential issue, which potentially results in suboptimal performance of STGNNs that are trained on a specific time period to accurately predict future unseen data.

Meanwhile, *dynamic spatial causation* is another essential nature of STG data that must be addressed for effective and unbiased representation learning in STG forecasting. While the majority of STGNNs

---

*Yuxuan Liang is the corresponding author of this paper. Email: yuxliang@outlook.com

rely on a distance-based adjacency matrix to perform message passing in the spatial domain [61, 26, 11], they lack adaptability to dynamic changes in the relationships between nodes. This matrix is also sometimes inaccurate, as two closely located nodes may not necessarily have causal relationships, e.g., nodes belonging to different traffic streams. As an alternative solution, the attention mechanism [14, 64, 9] calculates the dynamic spatial correlations between nodes adaptively based on their input features. However, they still fall short of capturing the *ripple effects of causal relations*. Similar to how node signals can propagate information across graphs over time, causal relations (perceived as edge signals) can also exhibit this effect. For example in Figure 1b, when an accident occurs between node A and B at $t = 1$, it directly reduces the causal relation 1. At $t = 2$, this effect propagates to other relations, such as weakening relation 2 and strengthening relation 3. This happens because the accident decreases the proportion of traffic flow from node B observed by node D, thus simultaneously increasing the proportion of traffic flow from node C observed by D.

In this paper, our goal is to concurrently tackle the temporal OoD issue and dynamic spatial causation via causal treatments [36]. Primarily, we present a *Structural Causal Model* (SCM) to gain a deeper understanding of the data generation process of STG data. Based on SCM, we subsequently propose to 1) utilize *back-door adjustment* to enhance the generalization capability for unseen data; 2) apply *front-door adjustment* along with an edge-level convolution operator

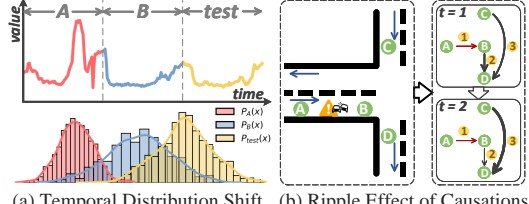

(a) Temporal Distribution Shift  (b) Ripple Effect of Causations

Figure 1: (a) Illustration of temporal OoD. (b) Spatial causal relationship in the traffic system.

to effectively capture the dynamic causation between nodes. Our contributions is outlined as follows:

- **A causal lens and treatment for STG data.** We propose a causal perspective to decipher the underlying mechanisms governing the data generation process of STG-structured data. Building upon the causal treatment, we devise a novel framework termed **Ca**usal **S**patio-**T**emporal neural networks (**CaST**) for more accurate and interpretable STG forecasting.

- **Back-door adjustment for handling temporal OoD.** We articulate that the temporal OoD arises from unobserved factors, referred to as *temporal environments*. Applying the back-door adjustment, we design a disentanglement block to separate the invariant part (we call it *entity*) and environments from input data. These environments are further discretized by vector quantization [46] which incorporates a learnable environment codebook. By assigning different weights to these environment vectors, our model can effectively generalize on OoD data from unseen environments.

- **Front-door adjustment for capturing dynamic spatial causation.** Adopting a distanced-based adjacency matrix to capture spatial information around a node (referred to as *spatial context*) could include spurious causation. However, stratifying this context is computationally intensive, making back-door adjustments impractical for spatial confounding. We thus utilize the front-door adjustment and introduce a surrogate to mimic node information filtered based on the actual causal part in the spatial context. To better model the ripple effect of causation, we propose a novel de-confounding module to generate surrogate representations via causal edge-level convolution.

- **Empirical evidence.** We conduct extensive experiments on three real-world datasets to validate the effectiveness and practicality of our model. The empirical results demonstrate that CaST not only outperforms existing methods consistently on these datasets, but can also be easily interpreted.

## 2 Related work

**Spatio-Temporal Graph Forecasting.** Recently, STG forecasting has garnered considerable attention, with numerous studies focusing on diverse aspects of this domain. Based on the foundation of GNNs [23], STGNNs [50, 18, 38, 20] have been developed to learn the spatio-temporal dependencies in STG data. By incorporating temporal components, such as TCN [2] or RNN [13], STGNNs are capable of modeling both spatial correlations and temporal dependencies in STG data. Pioneering examples include DCRNN [26], STGCN [61], and ST-MGCN [11]. Following these studies, Graph WaveNet [56] and AGCRN [1] leverage an adaptive adjacency matrix to improve the predictive performance. ASTGCN [14] and GMAN [64] utilize attention mechanisms to learn dynamic spatio-temporal dependencies within STG data. STGODE [8] and STGNCDE [5] capture the continuous spatial-temporal dynamics by using neural ordinary differential equations. ST-MetaNet [34, 35] and AutoSTG [33] exploit meta learning and AutoML for learning STGs, respectively. However, none of these approaches can simultaneously address the temporal OoD issue and dynamic spatial causation.

**Causal Inference.** Causal inference [36, 12] seeks to investigate causal relationships between variables, ensuring stable and robust learning and inference. Integrating deep learning techniques with causal inference has shown great promise in recent years, especially in computer vision [62, 51, 29], natural language processing [37, 44], and recommender system [65, 10]. However, in the field of STG forecasting, the application of causal inference is still in its infancy. Related methods like graph-based causal models are typically designed for graph/node classification tasks [43, 66] and link prediction [25]. For sequential data, causal inference is commonly used to address the temporal OoD issue by learning disentangled seasonal-trend representations [53] or environment-specific representations [57]. When adapting to STGs, these methods face hurdles, as graph-based models cannot tackle temporal OoD issues, while sequence-based models fail to accommodate spatial dependencies. In this study, we investigate the STG data generation process through a causal lens and employ causal techniques to mitigate confounding effects in both the temporal and spatial domains for STG forecasting.

## 3 Causal Interpretation for STG Data Generation

**Problem Statement.** We denote $X^t \in \mathbb{R}^{N \times D}$ as the signals of $N$ nodes at time step $t$, where each node has $D$ features. Given the historical signals from the previous $T$ steps, we aim to learn a function $\mathcal{F}(\cdot)$ that forecasts signals over

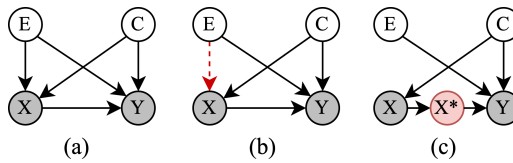

(a)          (b)          (c)

Figure 2: SCMs of (a) STG generation under real-world scenarios; (b) back-door adjustment for $E$; (c) front-door adjustment for $C$.

the next $S$ steps: $[X^{(t-T):t}] \overset{\mathcal{F}(\cdot)}{\to} [Y^{(t+1):(t+S)}]$, where $X^{(t-T):t} \in \mathbb{R}^{T \times N \times D}$, $Y^{(t+1):(t+S)} \in \mathbb{R}^{S \times N \times D'}$ and $D'$ is the output dimension. For conciseness, we refer to $X^{(t-T):t}$ as $X$ and $Y^{(t+1):(t+S)}$ as $Y$ in the rest of the paper.

**A Causal Look on STG.** From a causal standpoint, we construct a Structural Causal Model (SCM) [36] (see Figure 2a) to illustrate the causal relationships among four variables: temporal environment $E$, spatial context $C$, historical node signals $X$, and future signals $Y$. Arrows from one variable to another signify causal-effect relationships. For simplicity, we assume $E$ and $C$ are mutually independent. Based on the above definitions, the causal relationships in Figure 2a can be denoted as $P(X, Y | E, C) = P(X | E, C) P(Y | X, E, C)$. We detail these causal-effect relationships below:

- $X \leftarrow E \rightarrow Y$. The temporal OoD is an inherent property of STG data, e.g., $X$ and $Y$, where $P(X^t) \neq P(X^{t+\Delta t})$ at different time steps $t$ and $t + \Delta t$. This phenomenon can arise due to changes in external variables over time, which we refer to as temporal environments $E$ [57]. For example, external factors such as weather and events can significantly affect traffic flow observations.

- $X \leftarrow C \rightarrow Y$. The historical and future data $X$ and $Y$ are intrinsically affected by the encompassing spatial context $C$ surrounding a node. This influence, however, can comprise both spurious and genuine causal components. The spurious aspects may encompass nodes that exhibit either spatial distance or semantic similarity yet lack causal connections, as elucidated in Section 1.

- $X \rightarrow Y$. This relation is our primary goal established by the prediction model $Y = \mathcal{F}(X)$, which takes historical data $X$ as input and produces predictions for future node signals $Y$.

**Confounders and Causal Treatments.** Upon examining SCM, we observe two back-door paths between $X$ and $Y$, i.e., $X \leftarrow E \rightarrow Y$ and $X \leftarrow C \rightarrow Y$, where the temporal environment $E$ and spatial context $C$ act as confounding factors. This implies that some aspects of $X$, which are indicative of $Y$, are strongly impacted by $E$ and $C$. To mitigate the negative effect of the two confounders, we leverage the causal tools [12, 36] and *do-calculus* on variable $X$ to estimate $P(Y | do(X))$, where $do(\cdot)$ denotes the do-calculus. For the temporal OoD, we employ a popular de-confounding method called *back-door adjustment* [57, 43] to block the back-door path from $E$ to $X$ (the red dashed arrow in Figure 2b), so as to effectively remove $E$'s confounding effect. This necessitates implicit environment stratification (see Eq. 1). Spatial confounding, however, cannot be addressed by spatial context stratification, due to the computational burden engendered by the multitude of nodes, each exhibiting a unique contextual profile. Fortunately, the *front-door adjustment* allows us to introduce a mediating variable $X^*$ between $X$ and $Y$ to mimic a more accurate representation excluded the spurious parts in $C$ (the red node in Figure 2c) [54]. Note that we do not use the front-door adjustment to $E$ because this method mandates that the mediating variable is only affected by the cause variable and not by other confounding factors. While for temporal OoD scenarios, unseen future environments can be affected by time-varying factors, thus influencing the mediating variable. These two approaches effectively de-confound $E$ and $C$'s confounding effects, explained as follows.

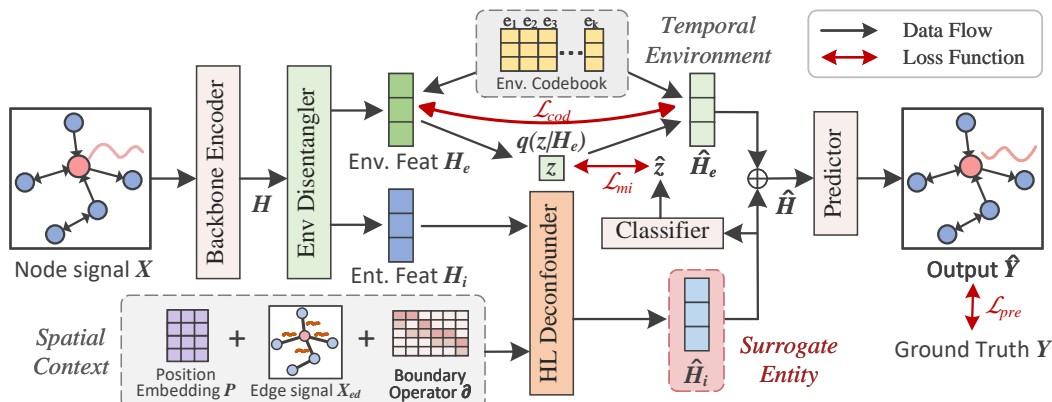

Figure 3: The pipeline of CaST. Env: Environment. Ent: Entity. Feat: Feature.

**Back-door Adjustment for** $E$. To forecast future time series $Y$ based on historical data $X$, it is imperative to address the confounding effect exerted by the *temporal confounder* $E$. To achieve this, we initially envisage a streamlined SCM, where $E$ constitutes the sole parents of $X$ (temporarily disregarding $C$) and employ back-door adjustment [36] to estimate $P(Y|do(X))$ by stratifying $E$ into discrete components $E = \{e_i\}_{i=1}^{|E|}$:

$$P(Y|do(X)) = \sum_{i=1}^{|E|} P(Y|X, E = e_i)P(E = e_i) \tag{1}$$

where the prior probability distribution of the environment confounder $P(E)$ is independent of $X$ and $Y$, allowing us to approximate the optimal scenario by enumerating $e_i$.

**Front-door Adjustment for** $C$. Once we have dealt with $E$, our next step is to de-confound the effect of spurious *spatial context* $C$ by using the front-door adjustment [12]. In the SCM depicted in Figure 2c, an instrumental variable $X^*$ is introduced between $X$ and $Y$ to mimic the node representation conditioned on their real causal relationships with other nodes. We then estimate the causal effect of $X$ on $Y$ as follows:

$$P(Y|do(X)) = \sum_{x^*} \sum_{x'} P(Y|X^* = x^*, X = x')P(X = x')P(X^* = x^*|X) \tag{2}$$

By observing $(X, X^*)$ pairs, we can estimate $P(Y|X^*, X)$[54]. This front-door adjustment provides a reliable estimation of the impact of $X$ on $Y$ while circumventing the confounding associations caused by $C$. We put the derivations of Eq. 1 and Eq. 2 in Appendix A.

## 4 Model Instantiations

We implement the above causal treatments by proposing a **Ca**usal **S**patio-**T**emporal neural network (**CaST**), as depicted in Figure 3. Our method takes historical observations $X$ as inputs to predict future signals $Y$. We will elaborate on the pipeline and each core component in the following parts.

**Back-door Adjustment** (see the top half of Figure 3). In order to attain Eq. 1, two steps need to be taken: (1) separating the environment feature from the input data, and (2) discretizing the environments. To accomplish this, we introduce an *Environment Disentangler* block, and a learnable *Environment Codebook* to obtain the desired stratification of environments.

**Front-door Adjustment** (see the bottom half of Figure 3). Obtaining $X^*$ and collecting the $(X, X^*)$ pairs to instantiate Eq. 2 is a non-trivial task that involves two main obstacles: (1) enumerating each spatial context, which can be computationally expensive, especially for large graphs, and (2) quantifying the causal effect of $X$ on $X^*$. To tackle these challenges, we introduce a block called Hodge-Laplacian (HL) Deconfounder, which is a neural topologically-based de-confounding module, to capture the dynamic causal relations of nodes as well as position embeddings to learn the nodes' global location information. With these two techniques, we can approximate the surrogate $X^*$.

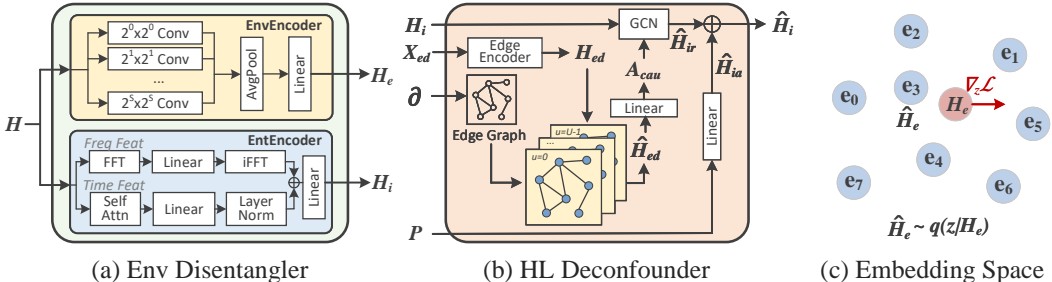

| (a) Env Disentangler | (b) HL Deconfounder | (c) Embedding Space |

Figure 4: (a) The structure of Env Disentangler. AvgPool: average pooling. Linear: linear projection. Attn: attention. FFT: Fast Fourier Transform. iFFT: inverse FFT. (b) The overview of HL Deconfounder. GCN: graph convolution. (c) The embedding space for the Environment Codebook. The output $\hat{H}_e$ is projected onto the closest vector $e_3$ and the gradient $\nabla_z \mathcal{L}$ pushes $H_e$ to change.

## 4.1 Temporal Environment Disentanglement

**Overview.** As shown in Figure 3, the input signals $X$ are first mapped to latent space as $H \in \mathbb{R}^{T \times N \times F}$ by a Backbone Encoder before entering the Env Disentangler, where $F$ means the hidden dimension. Then, this block separates $H$ into the environmental feature $H_e \in \mathbb{R}^{N \times F}$ and entity feature $H_i \in \mathbb{R}^{N \times F}$, analogous to background and foreground objects in the computer vision field [51]. Specifically, it captures environmental and entity information using two distinct components (see Figure 4a), including 1) EnvEncoder, which consists of a series of 1D convolutions, average pooling, and a linear projection; 2) EntEncoder, which extracts features from both time and frequency domains via Fast Fourier Transform and self-attention mechanism, respectively. The intuition of the block design of EnvEncoder and EntEncoder are discussed in Appendix B. After disentangled, for $H_e$, we compare it to an Environment Codebook and select the closest vector as the final representation $\hat{H}_e \in \mathbb{R}^{N \times F}$. The handling of $H_i$ will be explained in Section 4.2.

**Environment Codebook.** To stratify the environment $E$ in Eq. 1, we draw inspiration from [46] and develop a trainable environment codebook $e = \{e_1, e_2, \ldots, e_K\}$, which defines a latent embedding space $e \in \mathbb{R}^{K \times F}$. Here, $K$ signifies the discrete space size (i.e., the total number of environments), and $F$ denotes the dimension of each latent vector $e_i$. As depicted in Figure 3, after acquiring the environment representation $H_e$, we use a nearest neighbor look-up method in the shared embedding space $e$ to identify the closest latent vector for each node's environment representation. Given the environment feature of the $i$-th node $H_e(i) \in \mathbb{R}^F$, this process is calculated in the posterior categorical distribution $q(z_{ij} = k | H_e(i))$ as follows:

$$q(z_{ij} = k | H_e(i)) = \begin{cases} 1 & \text{for } k = \arg\min_j \|H_e(i) - e_j\|_2, \quad j \in \{1, 2, \ldots, K\}, \\ 0 & \text{otherwise.} \end{cases} \quad (3)$$

Once obtaining the latent variable $z \in \mathbb{R}^{N \times K}$, we derive the final environment representation $\hat{H}_e \in \mathbb{R}^{N \times F}$ by replacing each row in $H_e$ with its corresponding closest discrete vector in $e$. Note that this categorical probability in Eq. 3 is only used during the training process, whereas *a soft probability is used during testing to enable generalization to unseen environments*. This soft probability signifies the likelihood of environment representation for each node belonging to each environment, denoted as $\hat{H}_e(i) = \sum_{j=1}^{K} q(z_{ij}|H_e(i))e_j$, where $q(z_{ij}|H_e(i))$ ranges from 0 to 1. More discussion on how we achieve OoD generalization is provided in Appendix G.

**Representation Disentanglement.** We expect the environment and entity representations to be statistically independent, where entity representations carry minimal information (MI) about the environment. To achieve this, we employ an optimization objective inspired by Mutual Information Neural Estimation [3]. MI measures the information shared between $H_e$ and $H_i$, which is calculated using the Kullback-Leibler (KL) divergence between the joint probability $P(H_e; H_i)$ and the product of marginal distributions $P(H_e)P(H_i)$:

$$\mathcal{I}(H_e, H_i) = D_{KL}[P(H_e, H_i)\|P(H_e)P(H_i)]. \quad (4)$$

*By minimizing the mutual information, the overlap between $H_e$ and $H_i$ decreases. When it approaches zero, each representation is ensured to possess only self-contained information.* This approach transforms disentanglement into an optimization issue, which will be introduced in Section 4.3.

## 4.2 Spatial Context Filtering

**Overview.** Until now, we finished separating environment-entity and stratifying the environment $E$. We then shift our focus to the entity $H_i$. Our goal is to derive a surrogate $\hat{H}_i$ (i.e., the latent variable of $X^*$ in Figure 2c) that emulates a node representation containing only information propagated based on genuine causation within their spatial context. As emphasized in Section 1, it is essential to account for the ripple effects of causal relationships to accurately learn the surrogate. Thus the challenge is *how can we effectively model the ripple effect of dynamic causal relationships?* Since nodes' causal relations can naturally be regarded as edge features, an intuitive solution for this challenge is to *execute convolution operations on edges*. Inspired by [16], we build a higher-order graph over edges and use an edge-level spectral filter, i.e., the Hodge-Laplacian operator, to represent the propagation of causal relations. This forms the core of HL Deconfounder block (see Figure 4b).

Moreover, the locational information of nodes incorporates a more global spatial perspective, which can be seamlessly implemented using a position embedding [31]. We showcase the effects of it in Appendix F. Ultimately, the HL Deconfounder block ingests several inputs: the entity variable $H_i$, the edge signal $X_{ed} \in \mathbb{R}^{M \times F'}$, the boundary operator $\partial[24]$, and the position embedding $P \in \mathbb{R}^{N \times D_p}$, where $M$ and $F'$ mean the number of nodes and the dimension of their features in the built higher-order graph, respectively, and $D_p$ denote the embedding dimension. These inputs are processed to yield $\hat{H}_i \in \mathbb{R}^{N \times F}$. We then provide an exhaustive breakdown of this process, complemented with associated formulations.

**Edge Graph Construction.** We first construct a higher-order graph over edges by employing the boundary operator [24], which is a mathematical tool used in graph topology to connect different graph elements. Specifically, the first-order boundary operator $\partial_1$ maps pairs of nodes to edges, while the second-order boundary operator $\partial_2$ maps pairs of edges to triangles. Here we use $\partial_1$ and $\partial_2$ to establish a higher-order graph on edges, which facilitates subsequent convolution operations.

**Hodge-Laplacian Operator & Approximation.** With the edge graph, we can perform edge convolution to filter edge signals that contain genuine causation for a node. The Hodge-Laplacian (HL) operator [16, 39] is a spectral operator defined on the boundary operator. The first-order HL operator is defined as: $\mathbf{L} = \partial_2 \partial_2^\top + \partial_1^\top \partial_1$. Solving the eigensystem $\mathbf{L}\psi^j = \lambda^j \psi^j$ produces orthonormal bases $\{\psi^0, \psi^1, \psi^2, \dots\}$. The HL spectral filter $h$ with spectrum $h(\lambda)$ is defined as: $h(\cdot, \cdot) = \sum_{j=0}^{\infty} h(\lambda^j)\psi^j(\cdot)\psi^j(\cdot)$. To approximate $h(\lambda)$, we follow [16] and expand it as a series of Laguerre polynomials $T_u$ with learnable coefficients $\theta_u$:

$$h(\lambda) = \sum_{u=0}^{U-1} \theta_u T_u(\lambda), \tag{5}$$

where $T_u$ can be computed via the recurrence relation $T_{u+1}(\lambda) = \frac{(2u+1-\lambda)-uT_{u-1}(\lambda)}{u+1}$, with initial states $T_0(\lambda) = 1$ and $T_1(\lambda) = 1 - \lambda$. More details of these operators can be found in Appendix C.

**Causation Filtering & Surrogate Variable.** Shown in Figure 4b, the edge signal $X_{ed}$ is mapped into the latent space by the edge encoder to obtain $H_{ed} \in \mathbb{R}^{M \times F}$. Next, we acquire a new causal relations representation $\hat{H}_{ed} \in \mathbb{R}^{M \times F}$ by spectral filtering over $H_{ed}$: $\hat{H}_{ed} = h * H_{ed} = \sum_{u=0}^{U-1} \theta_u T_u(\mathbf{L})H_{ed}$. We then use a linear transformer to calculate the causal strength $A_{cau} \in \mathbb{R}^{N_e \times K_b}$ and derive $\hat{H}_{ir} \in \mathbb{R}^{N \times F}$ using Graph Convolutional Networks (GCN), where nodes' information is filtered by their genuine causation. $N_e$ and $K_b$ are the numbers of causal relations (i.e., edges) and the GCN block's depth. Meanwhile, a position embedding $P$ is used to generate $\hat{H}_{ia} \in \mathbb{R}^{N \times F}$ via a linear transform. We then obtain the surrogate $\hat{H}_i = \hat{H}_{ir} + \hat{H}_{ia}$. Ultimately, we concatenate it with $\hat{H}_e$ (obtained in Section 4.1) to form the predictor's input $\hat{H}$ and obtain the final prediction $\hat{Y}$.

## 4.3 Optimization

**Environment Codebook.** We train the codebook following [46]. During forward computation, the nearest embedding $\hat{H}_e$ is concatenated with the entity representation $\hat{H}_i$ and fed to the decoder (i.e., predictor). In the backward pass, the gradient $\nabla_z \mathcal{L}$ (see Figure 4c) is passed unaltered to the EnvEncoder within the Env Disentangler block. These gradients convey valuable information for adjusting the EnvEncoder's output to minimize the loss. The loss function has two components, namely the prediction loss $\mathcal{L}_{pre}$ and the codebook loss $\mathcal{L}_{cod}$:

$$\mathcal{L}_{pre} = -logP(Y|\hat{H}_e, \hat{H}_i), \qquad \mathcal{L}_{cod} = ||sg[H_e] - e||_2^2 + \alpha||H_e - sg[e]||_2^2 \tag{6}$$

where $\alpha$ is a balancing hyperparameter and $sg[\cdot]$ denotes the *stopgradient operator*, acting in a dual capacity – as an identity operator during forward computation, and has zero partial derivatives during the backward pass. As a result, it prevents its input from being updated. The predictor is optimized solely by $\mathcal{L}_{pre}$, whereas the EnvEncoder is optimized by both $\mathcal{L}_{pre}$ and the second term of $\mathcal{L}_{cod}$. The first term of $\mathcal{L}_{cod}$ optimizes the codebook.

**Mutual Information Regularization.** To minimize $\mathcal{I}(H_i, H_e)$, we use a classifier to predict $z$ in Eq. 3 based on $\hat{H}_i$, denoted as $\hat{z}$. The objective is to thwart the classifier to discern the true labels, or in other terms, to ensure that the classifier can not determine the true corresponding environment based on the information provided by $\hat{H}_i$. To achieve this, we introduce the MI loss $\mathcal{L}_{mi}$ that minimize the cross-entropy between $z$ and $\hat{z}$ to encourage $\hat{z}$ to move away from the true labels $z$ and towards a uniform distribution:

$$\mathcal{L}_{mi} = \mathcal{I}(H_i, H_e) = \sum\nolimits_{k=1}^{K} z^k \log\left(\hat{z}^k\right) \tag{7}$$

where $z^k$ and $\hat{z}^k$ means labels belonging to $e_k$. The overall loss function is obtained by combining these three losses: $\mathcal{L} = \mathcal{L}_{pre} + \mathcal{L}_{cod} + \beta\mathcal{L}_{mi}$, where $\beta$ regulates the trade-off of the MI loss.

## 5 Experiments

**Datasets & Baselines.** We conduct experiments using three real-world datasets (PEMS08 [42], AIR-BJ [59], and AIR-GZ [59]) from two distinct domains to evaluate our proposed method. PEMS08 contains the traffic flow data in San Bernardino from Jul. to Aug. in 2016, with 170 detectors on 8 roads with a time interval of 5 minutes. AIR-BJ and AIR-GZ contain one-year $PM_{2.5}$ readings collected from air quality monitoring stations in Beijing and Guangzhou, respectively. *Our task is to predict the next 24 steps based on the past 24 steps*. For comparison, we select two classical methods (HA [63] and VAR [45]) and seven state-of-the-art STGNNs for STG forecasting, including DCRNN [26], STGCN [61], ASTGCN[14], MTGNN[55], AGCRN[1], GMSDR [30] and STGNCDE [5]. More details about the datasets and baselines can be found in Appendix D and E, respectively.

**Implementation Details.** We implement CaST and baselines with PyTorch 1.10.2 on a server with NVIDIA RTX A6000. We use the TCN [2] as the backbone encoder and a 3-layer MLP as the predictor and the classifier. Our model is trained using Adam optimizer [22] with a learning rate of 0.001 and a batch size of 64. For the hidden dimension $F$, we conduct a grid search over $\{8, 16, 32, 64\}$. For the number of layers in each convolutional block, we test it from 1 to 3. The codebook size $K$ is searched over $\{5, 10, 20\}$. See the final setting of our model on each dataset in Appendix D.

### 5.1 Comparison to State-Of-The-Art Methods

We evaluate CaST and baselines in terms of Mean Absolute Error (MAE) and Root Mean Squared Error (RMSE), where lower metrics indicate better performance. Each method is executed five times, and we report the mean and standard deviation of both metrics for each model in Table 1. From this table, we have three key findings: 1) CaST clearly outperforms all competing baselines over the three datasets, whereas the second-best performing model is not consistent across all cases. This reveals that CaST demonstrates a more stable and reliable accuracy across various datasets, highlighting its versatility and adaptability to various domains. 2) STGNN-based models largely surpass conventional methods, i.e., HA and VAR, by virtue of their superior model capacity. 3) While baseline models such as AGCRN and MTGNN can achieve runner-up performance in certain cases, they exhibit a

Table 1: 5-run error comparison. The bold/underlined font means the best/the second-best result.

| Model | PEMS08 (24→24) | | AIR-BJ (24→24) | | AIR-GZ (24→24) | |
|---|---|---|---|---|---|---|
| | MAE | RMSE | MAE | RMSE | MAE | RMSE |
| HA(2017) | 58.83 | 81.96 | 32.12 | 43.95 | 19.56 | 25.77 |
| VAR(1991) | 37.04 | 53.08 | 29.79 | 42.04 | 14.97 | 20.61 |
| DCRNN(2017) | 22.10 ± 0.45 | 33.96 ± 0.59 | 23.72 ± 0.36 | 35.84 ± 0.56 | 12.99 ± 0.26 | 18.27 ± 0.41 |
| STGCN(2018) | 18.60 ± 0.08 | 28.44 ± 0.15 | 23.71 ± 0.21 | 36.30 ± 0.58 | 12.69 ± 0.04 | 17.66 ± 0.09 |
| ASTGCN(2019) | 20.36 ± 0.48 | 30.87 ± 0.55 | 23.78 ± 0.22 | 35.91 ± 0.11 | 12.91 ± 0.15 | 18.02 ± 0.27 |
| MTGNN(2020) | 18.13 ± 0.10 | 28.85 ± 0.12 | 24.35 ± 0.74 | 38.97 ± 1.81 | 12.43 ± 0.11 | 17.99 ± 0.18 |
| AGCRN(2020) | 17.06 ± 0.14 | 26.80 ± 0.15 | 23.43 ± 0.29 | 35.66 ± 0.57 | 12.74 ± 0.01 | 17.49 ± 0.01 |
| GMSDR(2022) | 18.34 ± 0.68 | 28.36 ± 1.01 | 25.92 ± 0.52 | 39.60 ± 0.44 | 13.47 ± 0.31 | 19.04 ± 0.46 |
| STGNCDE(2022) | 17.55 ± 0.30 | 27.28 ± 0.36 | 24.35 ± 0.31 | 35.91 ± 0.48 | 13.70 ± 0.10 | 19.15 ± 0.07 |
| CaST (ours) | **16.44** ± 0.10 | **26.61** ± 0.15 | **22.90** ± 0.09 | **34.84** ± 0.11 | **12.36** ± 0.01 | **17.25** ± 0.05 |

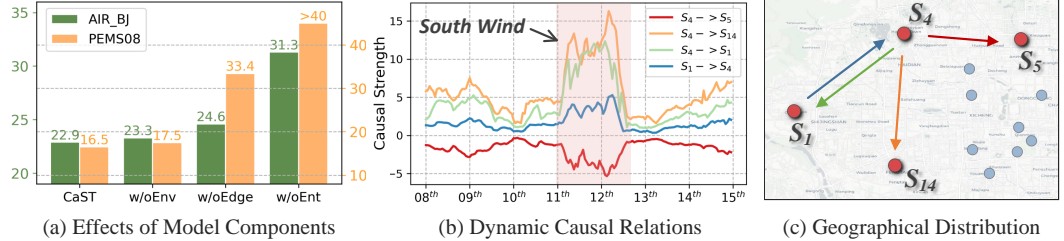

|  | (a) Effects of Model Components | (b) Dynamic Causal Relations | (c) Geographical Distribution |

Figure 5: (a) Effects of each core component on MAE. w/o: without. (b) Visualization of the dynamic causal relationships between nodes on AIR-BJ. (c) Distribution of air quality stations.

larger standard deviation compared to CaST. This demonstrates that CaST not only offers superior predictive accuracy but also showcases robustness and generalization capabilities. The evaluation of our proposed model confirms that incorporating causal tools not only enhances interpretability but also improves predictive accuracy and generalization performance across different scenarios. We also present the assessment of model performance on various future time steps in Appendix F.

## 5.2 Ablation Study & Interpretation Analysis

**Effects of Core Components.** To examine the effectiveness of each core component in our proposed model, we conducted an ablation study based on the following variants for comparison: a) **w/o Env**, which excludes environment features for prediction. b) **w/o Ent**, which omits entity features for prediction. c) **w/o Edge**, which does not utilize the causal score to guide the spatial message passing. The MAE results for two datasets, AIR-BJ and PEMS08, are displayed in Figure 5a. We observe that all components contribute to the model's performance. In addition, removing the entity component harms performance more than omitting the environment component, confirming our model's capacity to distinguish between the two. Moreover, removing the edge filtering affects PEMS08 more than AIR-BJ. This can be attributed to the unique properties of the down HL operator and the datasets, aligning better with the incompressible traffic flow of PEMS08 for causal capture, as opposed to AIR-BJ's non-divergence-free flow [40, 4]. A more detailed discussion can be found in Appendix G.

**Effects of Edge Convolution.** Our model utilizes a spectral filter on edges in the spatial de-confounding block, which enables it to capture the ripple effects of dynamic causal relationships. To validate the superiority of our edge convolution module over

Table 2: Variant results on MAE over AIR-BJ. s: steps.

| Variant | Overall | 1-8s | 9-16s | 17-24s |
|---------|---------|------|-------|--------|
| CaST-ADP | 24.28 | 16.42 | 26.06 | 30.36 |
| CaST-GAT | 23.77 | 14.76 | 25.75 | 30.80 |
| CaST | **22.90** | **13.79** | **24.86** | **30.05** |

existing spatial learning methods, we conduct an ablation study on the AIR-BJ dataset by comparing the performance of CaST against two variants: **CaST-ADP** which replaces the edge convolution with a self-adaptive adjacency matrix [56], and **CaST-GAT** which employs the graph attention mechanism to obtain the causal score [47]. The results presented in Table 2 indicate that our edge convolution can adeptly discern the causal strengths between nodes, thereby resulting in enhanced performance.

**Visualization of Dynamic Spatial Causation.** To show the power of edge convolutions, we depict the trend of learned causal relations among four air quality stations in Beijing. Note that we do not incorporate any external features (e.g., weather and wind) as input. Considering the fact that the dispersion of air quality is strongly associated with wind direction, we select a one-week period in Nov. 2019, during which a *south wind* occurred on 11th Nov. The varying causal relations among the selected stations and their geolocations are displayed in Figures 5b and 5c. From them, three key observations can be obtained. **Obs1**: The causal relationships $S_4 \to S_{14}$ and $S_4 \to S_1$ exhibit similar patterns when there is a south wind, as $S_1$ and $S_{14}$ are both located south of $S_4$, which is reasonable to suggest that the causal relationship would be stronger in this case. **Obs2**: $S_4 \to S_5$ shows an opposite changing direction compared to the former two relationships, as south winds carry $PM_{2.5}$ southward, resulting in the causal strength from $S_4$ to eastern stations like $S_5$. **Obs3**: $S_4 \to S_1$ and $S_1 \to S_4$ display distinct patterns of change. While the causal strength from $S_4$ to $S_1$ increases with the occurrence of a south wind, that from $S_1$ to $S_4$ shows no significant change. The reason is that a south wind carries $PM_{2.5}$ towards the south, increasing its concentration in $S_1$ and strengthening the causal relationship from $S_4$ to $S_1$. However, the wind's effect is weaker in the opposite direction, leading to no significant changes in the causal strength from $S_1$ to $S_4$. These findings underscore the capability of CaST to effectively capture the dynamic causal relationships among nodes.

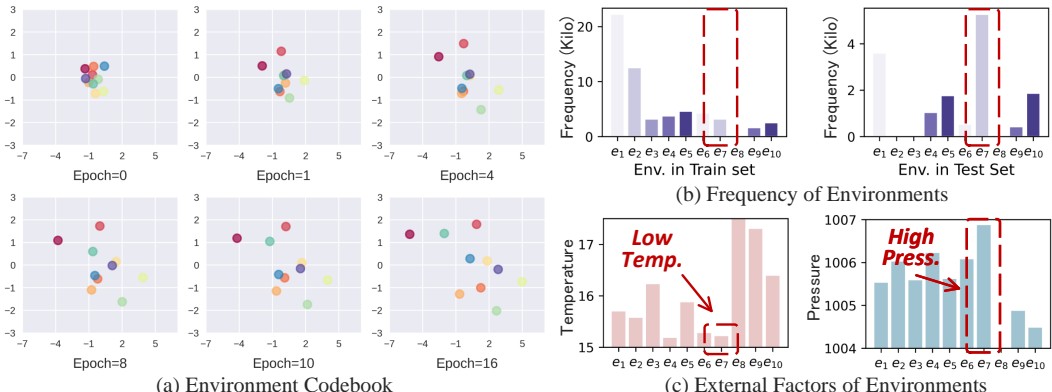

(a) Environment Codebook

(b) Frequency of Environments

(c) External Factors of Environments

Figure 7: Case study on AIR-BJ (better viewed in color). (a) Environment codebook at different training epochs by PCA. (b) Distribution of environments for the station $S_{31}$ in the train and test sets. Env: Environment. (c) The mean of temperature (Temp) and pressure (Press) in each environment.

**Analysis on Environmental Codebook.** We employ the environmental codebook as a latent representation to address the temporal OoD issue. In Figure 7a, we visualize the environmental codebook vectors using PCA dimensional reduction with $K = 10$ on AIR-BJ at different training epochs. Each color corresponds to a particular environment's embedding. Remarkably, starting from the same initialized positions, the

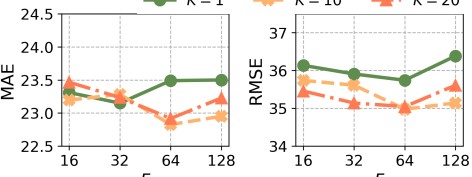

Figure 6: Effects of $K$ and $F$ on AIR-BJ.

environment embeddings gradually diverge and move in distinct directions. Next, we assess the influence of several critical hyperparameters in CaST, including the hidden dimension $F$ and the environment codebook size $K$. We examine how CaST performs on AIR-BJ while varying these hyperparameters and display the results in Figure 6. The empirical results indicate that the model's accuracy exhibits a lower sensitivity to the hidden size $F$. When the hidden size is small (i.e., $F < 32$), the performance is relatively unaffected by the choice of $K$. Consequently, the model is less reliant on the granularity of the environment representation, as provided by $K$, when the hidden size is small.

**Interpretation of Temporal Environments.** To further investigate what the codebook has learned, we visualize the environment types distribution of station $S_{31}$ in the training set (Jan. - Aug. 2019) and testing set (Nov. - Dec. 2019) in Figure 7b. Note that the number of environments in the test set is computed as a sum of probabilities since we use the likelihood for each environment to generate unseen environments during the test phase. A substantial discrepancy in environmental distribution between the two sets is observed, with a significant increase in the occurrence frequency of $e_7$ in the testing set compared to the training set, indicating the existence of a temporal distribution shift. As previously discussed, temporal environments are associated with related external factors that change over time. We then collect meteorological features in Beijing for 2019 and calculate the mean of two related variables, i.e., temperature and pressure, for each environment (see Figure 7c). We find that $e_7$ is characterized by low temperature and high pressure, which is consistent with the fact that Nov. - Dec. typically exhibit lower temperatures and higher pressures compared to Jan. - Oct. The distribution of these variables in accordingly periods is presented in Appendix F. This highlights our model's capability to learn informative environment representations.

## 6 Conclusion

In this paper, we present the first attempt to jointly address the challenges of tackling the temporal OoD issue and modeling dynamic spatial causation in the STG forecasting task from a causal perspective. Building upon a structural causal model, we present a causal spatio-temproal neural network termed CaST that performs the back-door adjustment and front-door adjustment to resolve the two challenges, respectively. Extensive experiments over three datasets can verify the effectiveness, generalizability, and interpretability of our model. We provide more discussions in Appendix G, including the limitations of our model, potential future directions, and its social impacts.

## Acknowledgments and Disclosure of Funding

This research is supported by Singapore Ministry of Education Academic Research Fund Tier 2 under MOE's official grant number T2EP20221-0023. This work is also supported by Guangzhou Municiple Science and Technology Project 2023A03J0011. We sincerely thank all reviewers for their insightful and constructive comments in improving this paper.

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

## A  Derivation of Back-door and Front-door Adjustment

We first introduce three basic rules of do-calculus $do(\cdot)$, then we present the derivation of back-door and front-door adjustment for the proposed SCM in Figure 2a based on these rules [36, 12].

**Rules of Do-calculus.** Let $G$ be a directed acyclic graph with three nodes: $X$, $Y$, and $Z$. Denote the *interventional graph* as $G_{do(X)}$, which is identical to $G$ except for the removal of all arrows leading to $X$ from its parents. Denote the *nullified graph* as $G_{null(X)}$, where all arrows from $X$ have been removed. For better understanding, we illustrate $G$, $G_{do(X)}$ and $G_{null(X)}$ in Figure 8.

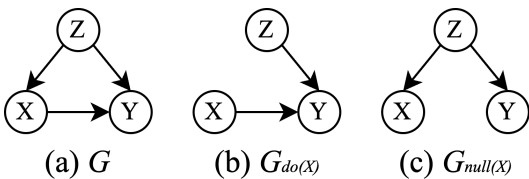

Figure 8: Illustration of (a) directed acyclic graph, (b) interventional graph, and (c) nullified graph.

Based on these definitions, we can introduce the following three rules of do-calculus[12]:

- **Rule 1**: *Insertion/deletion of observations*

$$P(y|do(x), z) = P(y|do(x)), \quad \text{if } (y \perp\!\!\!\perp z|x)_{G_{do(x)}}$$

- **Rule 2**: *Action/observation exchange*

$$P(y|do(x), do(z)) = P(y|do(x), z), \quad \text{if } (y \perp\!\!\!\perp z|x)_{G_{do(x),null(z)}}$$

- **Rule 3**: *Insertion/deletion of actions*

$$P(y|do(x), do(z)) = P(y|do(x)), \quad \text{if } (y \perp\!\!\!\perp z|x)_{G_{do(x),do(z)}}$$

where $(y \perp\!\!\!\perp z|x)_G$ means that that $y$ and $z$ are independent of each other given $x$ in $G$. For example, *Rule 1* asserts that variables $z$ can be removed from the conditioning set if the intervention variables $x$ d-separate $y$ from $z$ in the intervention graph $G_{do(x)}$. Note that Figure 8 is an illustration of the definitions of $G$ and its interventional and nullified counterparts, not mandatory for them to comply with these rules.

After introducing the definition, we shift our focus back to Figure 2 and apply the back-door and front-door adjustment on $E$ and $C$, respectively.

**Back-door Adjustment.** We initially envisage a streamlined directed acyclic graph, where the environment $E$ constitutes the sole parents of $X$ (temporarily disregarding $C$). By applying the rules of do-calculus, we can derive the back-door criterion as follows:

$$P(Y|do(X)) = \sum_e P(Y|do(X), E = e)P(E = e|do(X)) \quad \textit{(Bayes Rule)}$$

$$= \sum_e P(Y|do(X), E = e)P(E = e) \quad \textit{(Rule 3)}$$

$$= \sum_e P(Y|X, E = e)P(E = e) \quad \textit{(Rule 2)}$$

By stratifying $E$ into discrete components $E = \{e_i\}_{i=1}^{|E|}$, we can express the following:

$$P(Y|do(X)) = \sum_{i=1}^{|E|} P(Y|X, E = e_i)P(E = e_i) \tag{8}$$

**Front-door Adjustment.** After de-confounding $E$, we have $G_{null(E)}$. Then, we turn our attention to the spatial confounder $C$ via front-door adjustment as follows:

$$P(Y|do(X)) = \sum_{x^*} P(Y|do(X^* = x^*))P(X^* = x^*|do(X)) \quad \textit{(Bayes Rule)}$$

$$= \sum_{x^*} \sum_{x'} P(Y|X^* = x^*, X = x')P(X = x')P(X^* = x^*|do(X)) \quad \textit{(Bayes Rule)}$$

$$= \sum_{x^*} \sum_{x'} P(X^* = x^*|X)P(Y|X^* = x^*, X = x')P(X = x') \quad \textit{(Rule 2)}$$

# B  More Details of Env Disentangler

We devise the Env Disentagneler to capture the environmental and entity representations, as depicted in Figure 4a. The Env Disentangler comprises two distinct components: the EnvEncoder, which is responsible for capturing environmental information, and the EntEncoder, which focuses on extracting entity features. In the rest of this section, we will provide a detailed explanation of these components and their respective roles in our model.

**EnvEncoder.** We design the EnvEncoder (illustrated by the yellow block) to extract environment features. As we have previously discussed in Section 3, environmental factors can be subtle and often exhibit long-term changes, such as seasonal variations or policy shifts, our aim is to capture more *global* information from an input time series to better represent these underlying environmental influences. The EnvEncoder is comprised of a mixture of 1D convolutions with a kernel size of $2^i, i \in \{0, 1, \ldots, S_k\}$, where $S_k$ is a hyperparameter. These convolutions are followed by an average pooling layer and a linear projection to obtain the environment representation $H_e$.

**EntEncoder.** We devise the EntEncoder (illustrated by the blue block) to extract entity information. As entities often contain more *local* and *periodic* information, we learn their representations from two aspects: the frequency domain (upper line) and the time domain (lower line). In the EntEncoder's upper line, we first perform a Fast Fourier Transform (FFT) operation on the input data and then feed it into a linear layer to capture the frequency information. Subsequently, we apply the inverse Fast Fourier Transform (iFFT) to return it to the time domain. In the lower line of the EntEncoder, we utilize the self-attention mechanism to capture the intricate relationships between different time steps, thereby focusing on crucial aspects of the time domain information. This is followed by a layer normalization. By combining the frequency and time domain information, and feeding it into a linear layer, we can obtain the entity features $H_i$.

# C  More Details of HL Deconfounder

The HL Deconfounder is developed (see Figure 4b) to learn a surrogate that mimics the node information filtered by the dynamic causal score. To achieve an accurate representation learning for this surrogate, we use an edge-level convolution to model the ripple effect of the causal relationships, which is mathematically based on the boundary operator [24] and Hodge-Laplacian operator [16]. We provide details on these two techniques in the following part.

**Boundary operator.** Graphs consist of nodes and edges, which can be mathematically represented as 0-dimensional and 1-dimensional simplices, respectively [7]. The topology of a graph can be described by a boundary operator $\partial_k$ [24], where $\partial_1$ encodes the connections between two 0-dimensional simplices (nodes, denoted as $\sigma_0$) to form a 1-dimensional simplex (edge, denoted as $\sigma_1$). The second order boundary operator $\partial_2$ represents the connections between 1-dimensional simplices (edges) to form 2-dimensional simplices (triangles, denoted as $\sigma_2$). The boundary operator $\partial_k$ can be formulated in the matrix form as follows:

$$[\partial_k]_{ij} = \begin{cases} 1 & \text{if } \sigma_{k-1}^i \text{ is positively oriented w.r.t. } \sigma_k^j, \\ -1 & \text{if } \sigma_{k-1}^i \text{ is negatively oriented w.r.t. } \sigma_k^j, \\ 0 & \text{otherwise} \end{cases}$$

where $\sigma_k^i, \sigma_k^j \in C_k$; $C_k = \{\sigma_k^1, \ldots, \sigma_k^q\}$ denotes the chain complex and $q$ is the number of simplices. For better understanding, we illustrate an example of the first-order boundary operator $\partial_1$ in Figure 9.

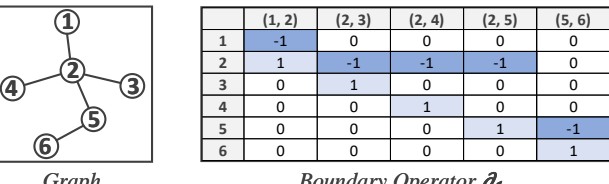

Figure 9: Example of the first boundary operator $\partial_1$.

**Hodge-Laplacian operator.** The Hodge-Laplacian (HL) operator[16] is proposed to achieve node-level and edge-level representation learning based on the features of neighboring nodes and edges. The boundary operators $\partial_k$ play a key role in achieving this goal and can be incorporated into the $k$-th Hodge-Laplacian operator, defined as:

$$\mathcal{L}_k = \partial_{k+1}\partial_{k+1}^\top + \partial_k^\top\partial_k, \tag{9}$$

where when $k = 0$, $\mathcal{L}_0 = \partial_1\partial_1^\top$ operates over nodes and is equivalent to the Graph Laplacian. When $k = 1$, the HL operator $\mathcal{L}_1 = \partial_2\partial_2^\top + \partial_1^\top\partial_1$ operates over edges. In this study, we use the $\mathcal{L}_1$ to achieve the edge-level convolution.

## D    Dataset and Experiment Settings

**Datasets.** Our experimental design involved selecting three real-world datasets from two distinct domains. The first dataset, PEMS08 [42], contains traffic flow data that was collected by sensors deployed on the road network. Traffic flow data is often considered to be a complex and challenging type of spatio-temporal data influenced by numerous factors, such as weather, time of day, and road conditions. The PEMS08 dataset is thus an ideal choice for our study as it allows us to examine the effectiveness of our proposed method in the context of a real-world traffic flow analysis scenario. The remaining two datasets, AIR-BJ and AIR-GZ [59] contain one-year PM$_{2.5}$ readings obtained from air quality monitoring stations located in Beijing and Guangzhou, respectively. The issue of air pollution is a major concern in many cities around the world, and the AIR-BJ and AIR-GZ datasets offered us the opportunity to investigate the application of our proposed method to spatio-temporal data associated with environmental monitoring. To provide a comprehensive overview of the characteristics of the datasets used in the experiments, we present their statistics in Table 3.

Table 3: Statistics of datasets.

| Dataset | #Nodes | #Edges | Data Type | Time interval | Date Range | #Samples | Train:Val:Test |
|---------|--------|--------|-----------|---------------|------------|----------|----------------|
| AIR-BJ | 34 | 82 | PM$_{2.5}$ | 1 hour | Jan.1, 2019 - Dec. 31, 2019 | 8,760 | 4:1:1 |
| AIR-GZ | 41 | 77 | PM$_{2.5}$ | 1 hour | Jan.1, 2017 - Dec. 31, 2017 | 8,760 | 4:1:1 |
| PEMS08 | 170 | 303 | Traffic flow | 5 minutes | Jul. 1, 2016 - Aug.31, 2016 | 17,856 | 8:1:1 |

**Edge Features.** Edge features are built for causality extraction. For each edge, we create three types of features: (1) Euclidean-based similarity $W_{ij}$, (2) Pearson correlations $\rho_{ij}$, and (3) Time-delayed Dynamic Time Warping (DTW) $R_{ij}$. The first two are static values, while the third one is a dynamic variable that changes over time.

- We compute the Euclidean distances between stations and construct a weighted adjacency matrix using thresholded Gaussian kernel [41] as follows:

$$W_{ij} = \begin{cases} \exp\left(-\frac{\text{dist}(v_i,v_j)^2}{\sigma^2}\right), & \text{if dist}(v_i, v_j) \leq \kappa \\ 0, & \text{otherwise} \end{cases}$$

  where $W_{ij} \in [0, 1]$ denotes the edge weight between node $v_i$ and node $v_j$; $\text{dist}(v_i, v_j)$ represents the distance between nodes $v_i$ and $v_j$; $\sigma$ is used as a standard deviation of distances and $\kappa$ is a threshold to exclude nodes that are considered to be very distant.

- The Pearson correlation coefficient measures the linear relationship between two signals generated by $i$-th node and $j$-th node, represented mathematically as:

$$\rho_{ij} = \frac{\text{cov}(X_i, X_j)}{\sigma_{X_i}\sigma_{X_j}},$$

  which ranges from -1 to 1, where 1 indicates a perfect positive linear relationship, 0 indicates no linear relationship, and -1 indicates a perfect negative linear relationship. $\text{cov}(\cdot, \cdot)$ represents the covariance between two random variables.

- Finally, we create a time-delay DTW $R_{ij}(\tau)$ with delay window size $\tau$ between the source node's signal $X_i^t$ and the target node's $\tau$ lag series $X_j^t$ [32]:

$$R_{ij}^\alpha(\tau) = DTW(X_i^t, X_j^{t-\alpha\tau}), \quad R_{ij}(\tau) = [R_{ij}^1(\tau), R_{ij}^2(\tau), \ldots, R_{ij}^N(\tau)]$$

  where $N = T/\tau$, $T$ refers the length of the input series and $\alpha \in \{1, 2, \ldots, N\}$. In this work, we set $\tau = 6$ and $T = 24$, resulting in 4 dimensions of $R_{ij}$ for each edge.

**Computation of HL Operator.** The computation of the first-order HL operator involves the calculation of $\partial_2$, which characterizes the interaction of edges and triangles. However, for the purpose of STG forecasting in this work, we simplify the computation by disregarding triangles and thus use the down HL operator (i.e., setting $\partial_2$ to zero) [39]. While in other applications where triangles or higher-order elements have explicit meanings, such as the Benzene Rings of Protein Structures in Molecular Structure Modeling, $\partial_2$ should be carefully defined.

**Evaluation Metrics.** We leverage Mean Absolute Error (MAE) and Root Mean Squared Error (RMSE) to evaluate the performance of models. Let $Y_i$ be the label of each entry of $Y$, and $\hat{Y}_i$ denote the corresponding prediction result, two metrics are calculated as follows:

$$\text{MAE}(Y, \hat{Y}) = \frac{1}{|Y|} \sum\nolimits_{i=1}^{|Y|} \left| Y_i - \hat{Y}_i \right|, \tag{10}$$

$$\text{RMSE}(Y, \hat{Y}) = \sqrt{\frac{1}{|Y|} \sum\nolimits_{i=1}^{|Y|} \left( Y_i - \hat{Y}_i \right)^2}, \tag{11}$$

**Final Settings of CaST.** We introduce the best hyperparameter configurations for each dataset as follows.

- For the PEMS08 dataset, the number of hidden units is set at 64, the codebook size $K$ is 20, and the number of layers in all convolutional blocks is 2. The position embedding size is set to 5, and the balancing coefficient for the mutual information loss, $\alpha$, is set to 0.5.

- For the AIR-BJ dataset, we configure the model with the following settings: the number of hidden units is set to 64, the codebook size $K$ is 10, and the number of layers for all convolutional blocks is 3. The position embedding size is set to 10, and the balancing coefficient for the mutual information loss, $\alpha$, is set to 1.

- For the AIR-GZ dataset, we use these configurations: the number of hidden units is set at 32, the codebook size $K$ equals 3, and the number of layers in all convolutional blocks is 3. The position embedding size is set to 5, and the balancing coefficient for the mutual information loss, $\alpha$, is set to 0.5.

# E   Details of Baselines

**Description & Settings.** We opted to include a selection of widely-used traditional methods and popular cutting-edge methods for comparative evaluation. We describe these baselines and outline the setting used in our experiments as follows. The number of parameters of each model is illustrated in Table 4.

- **HA** [63] forecasts future time series values by calculating the average of historical readings for the corresponding time periods.

- **VAR** [45] is a popular time series forecasting method that models the interdependence between multiple time series variables.

- **DCRNN** [26] is a type of neural network that uses diffusion convolution and sequence-to-sequence architecture to learn spatial dependencies and temporal relations. We utilize the source code provided on GitHub[2]. For each of the three datasets, we configure the model with 2 DCRNN layers and 64 hidden units. The receptive field in diffusion convolution is set to 3.

- **STGCN** [61] is a spatial-temporal graph convolution network that combines spectral graph convolution with 1D convolution to capture correlations between space and time. We employ the implementation shared by the authors on GitHub[3]. For each of the three datasets, we configure the channels of the three layers in the ST-Conv block to 64. We set the graph convolution kernel size $K$ and temporal convolution kernel size $K_t$ both to 3.

- **ASTGCN** [14] leverages an attention-based mechanism and a spatial-temporal convolution system to dynamically capture spatial-temporal correlations and model various temporal properties of

---

[2]https://github.com/liyaguang/DCRNN
[3]https://github.com/PKUAI26/STGCN-IJCAI-18

traffic flows, thereby providing superior traffic flow forecasting. We use the code released by the author[4] to implement this model. We set the number of blocks to 2 and the hidden size to 64.

- **MTGNN** [55] models the temporal dynamics of graph-structured data by aggregating information from spatially neighboring nodes and past time steps using a message-passing framework. We employ the official MTGNN implementation[5]. For all three datasets, we configure each MTGNN block with a graph convolution module and a temporal convolution module. The hidden dimension for each node in these modules is set to 40. Additionally, we set the convolution channel to 32 and the graph convolution depth to 2.

- **AGCRN** [1] uses Node Adaptive Parameter Learning and Data Adaptive Graph Generation modules to automatically infer inter-dependencies in traffic series and capture node-specific patterns. We use the code offered in Github[6] to run the experiments of this model. We set the order of Chebyshev polynomials to 2 and the hidden size to 32.

- **GMSDR** [30] improves upon RNNs by incorporating the hidden states of multiple historical time steps as input at each time unit. We use the available GMSDR code[7] for our experiments. Across all three datasets, we configure the model with 2 RNN layers and 64 hidden units.

- **STGNCDE** [5] is a spatio-temporal graph neural controlled differential equation model that uses two neural control differential equations to process spatial and sequential data. We employ the provided STGNCDE code[8] for our experiments. Across all three datasets, we configure the model with a node embedding size of 10 and a hidden vector dimensionality of 32.

Table 4: The number of parameters of models. The magnitude is Kilo.

| Model | AIR-BJ | AIR-GZ | PEMS08 |
|---|---|---|---|
| DCRNN | 94 | 94 | 94 |
| STGCN | 163 | 163 | 163 |
| ASTGNC | 90 | 92 | 210 |
| MTGNN | 155 | 156 | 166 |
| AGCRN | 189 | 189 | 189 |
| GMSDR | 229 | 246 | 553 |
| STGNCDE | 372 | 372 | 372 |
| CaST (ours) | 350 | 212 | 391 |

# F   More Experiments Results

**Experiments Results for Various Steps.** We performed five runs for each model, and Table 5 presents the mean and standard deviation of the results for different future time steps on the PEMS08 dataset. We observe that CaST outperforms the majority of compared models.

**Effects of Position Embedding.** As delineated in Section 4, we incorporate a position embedding $P$ alongside the edge signal to learn global location information. We execute our proposed model five times, both with and without $P$, for 4,000 iterations and present the loss curves for the training and evaluation sets in Figure 10. Losses are documented at 100-iteration intervals. Solid lines signify the mean loss across five runs, while the filled area indicates the standard deviation. For enhanced visualization, we employ a moving average method with a window size of 5 to smoothen the curves. Our findings suggest that models lacking position embedding may be susceptible to overfitting, as evidenced by the decreasing losses on the training set beyond 3,000 iterations, while losses on the evaluation set increase.

**Effects of Loss Balance.** We investigated the influence of different weightings for terms $\alpha$ and $\beta$ in our loss function on prediction performance, shown in Figure 11. The color intensity represents the value of the metrics, with lighter colors indicating better performances in MAE or RMSE. From the results, the combination of $\alpha = 0.5$ and $\beta = 1.5$ yielded the lowest MAE and RMSE, indicating this

---

[4]https://github.com/wanhuaiyu/ASTGCN
[5]https://github.com/nnzhan/MTGNN
[6]https://github.com/LeiBAI/AGCRN
[7]https://github.com/dcliu99/MSDR
[8]https://github.com/jeongwhanchoi/STG-NCDE

Table 5: 5-run results on PEMS08 for different time steps prediction. The bold/underlined font means the best/the second best result.

| Model | 1 - 8 steps | | 9 - 16 steps | | 17 - 24 steps | |
|---|---|---|---|---|---|---|
| | MAE | RMSE | MAE | RMSE | MAE | RMSE |
| HA | 58.60 | 81.60 | 58.85 | 82.02 | 59.04 | 82.27 |
| VAR | 21.07 | 32.05 | 37.10 | 53.85 | 52.94 | 74.58 |
| DCRNN | 16.48 ± 0.1 | 25.86 ± 0.17 | 22.18 ± 0.43 | 34.32 ± 0.59 | 27.63 ± 0.82 | 42.00 ± 1.07 |
| STGCN | 15.13 ± 0.05 | 23.47 ± 0.08 | 18.60 ± 0.09 | 28.55 ± 0.16 | 22.07 ± 0.10 | 33.45 ± 0.24 |
| ASTGCN | 16.27 ± 0.16 | 25.24 ± 0.22 | 20.38 ± 0.47 | 31.04 ± 0.60 | 24.43 ± 0.87 | 36.53 ± 0.95 |
| MTGNN | 14.80 ± 0.05 | 23.55 ± 0.06 | 18.12 ± 0.10 | 28.98 ± 0.12 | 21.47 ± 0.17 | 34.18 ± 0.18 |
| AGCRN | 15.19 ± 0.10 | 23.68 ± 0.14 | 17.24 ± 0.13 | 27.18 ± 0.14 | 18.74 ± 0.20 | 29.6 ± 0.19 |
| GMSDR | 18.04 ± 0.58 | 27.67 ± 0.80 | 18.50 ± 0.76 | 28.40 ± 1.13 | 18.49 ± 0.71 | 29.00 ± 1.13 |
| STGNCDE | 15.74 ± 0.30 | 24.29 ± 0.31 | 17.66 ± 0.30 | 27.46 ± 0.34 | 19.27 ± 0.34 | 30.15 ± 0.48 |
| CaST (ours) | 14.39 ± 0.04 | 22.83 ± 0.10 | 16.47 ± 0.08 | 26.80 ± 0.22 | 18.47 ± 0.20 | 30.28 ± 0.17 |

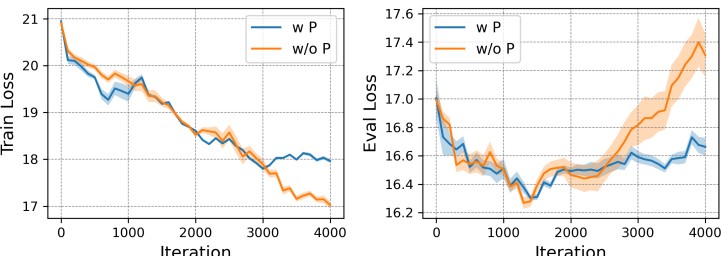

Figure 10: Training losses and evaluation losses on AIR-BJ. w: with. w/o: without.

balance might be optimal for minimizing the average error magnitude and reducing the impact of larger errors. However, it's noteworthy that the trends in performance improvement are not consistent, implying that the relationship between the terms in the loss function and the prediction performance could be complex and not straightforward.

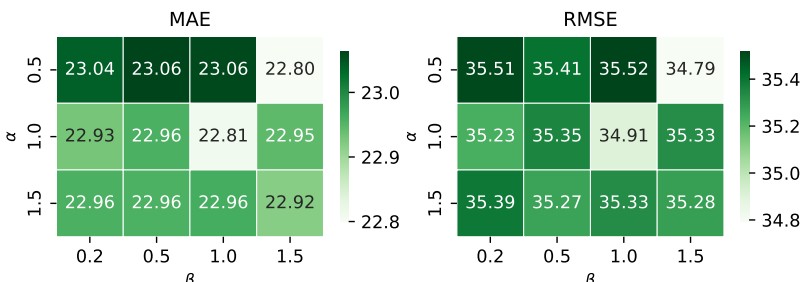

Figure 11: Heatmap of MAE and RMSE values for varying loss function weights on AIR-BJ dataset.

**External Factors Distribution.** The distribution of temperature and pressure in Beijing during the training period (January to August 2019) and the testing period (November to December 2019) is shown in Figure 12. The plot reveals that the testing period has a lower temperature and higher pressure than the training period, which is consistent with the observation in the

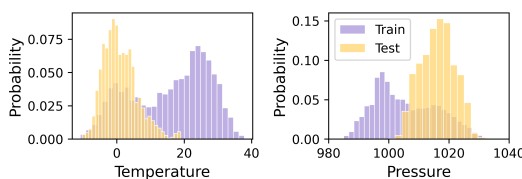

Figure 12: Distribution of external features.

case study of temporal environments visualization in Section 5.2 that the occurrence frequency of a specific temporal environment is higher in the testing set than in the training set. Note that these external factors are just used to do the interpretation analysis. When training the model, we did not use these external factors.

# G  More Discussions

**Generalization to the Unseen Environment.** We utilize the environmental codebook as a latent representation to capture information for each temporal environment type, whose goal is to represent unique and informative environments in the real world. During training, each environment is assigned an embedding from the codebook, while during testing, a soft probability is assigned to indicate the likelihood of the example belonging to each environment. This approach enables the generation of embeddings for unseen environments. For instance, if we have embeddings for high temperature and low pressure (i.e., $e_1$) and low temperature and high pressure (i.e., $e_2$), and we encounter a moderate environment with middle temperature and pressure, we can use different coefficients to generate a representative embedding $\lambda_1 e_1 + \lambda_2 e_2$ for this unseen environment.

**Divergence-free Property of Datasets.** In our ablation study on the effects of core components (see Figure 5), the impact of removing the edge filtering module significantly affects performance on the PEMS08 dataset, less so for AIR-BJ. This can be attributed to the unique properties of the down HL operator and the datasets. In the case of incompressible traffic flow, which is naturally divergence-free [40, 4], our HL deconfounder is able to capture significant causal information via the edge signals. This indeed might explain the significant drop in performance when edge signals are removed in the ablation study on the PEMS08 dataset. Conversely, for data types where the flow is not divergence-free, like the PM2.5 of the AIR-BJ dataset, the HL deconfounder may not capture as much information, leading to less significant performance drops when edge signals are removed.

**Complexity Analysis.** We omit hidden dimensionality in the following analysis for simplicity. The complexity of the spatial module is $O(M^2)$ induced by the edge convolution operation (i.e., HL Deconfounder) [21], with $M$ denoting the number of nodes; the complexity of the temporal module (i.e., Env Disentangler) is $O(T^2)$, where $T$ represents the historical signal length. Generally, we have $T \ll M$, leading to $O(M^2)$ additional overheads over the original Backbone Encoder.

**Limitations and Future Directions.** One limitation of our proposed framework is that it requires the stratification of temporal environments, which can potentially be infinite. This could pose a challenge in real-world applications where the environmental contexts may be continuously changing and difficult to categorize into a finite set of environments. In the future, we may explore the possibility of modeling continuous environments. Besides, while we have underscored the efficacy of the edge convolution CaST in discerning the causative relations between nodes, its computational speed lags relative to alternative methods, such as the adaptive adjacency matrix [56], due to the incorporation of edge graph construction, thereby increasing computational demands. However, given it can be done in the preprocessing stage in one shot, it won't produce extra computational costs in the training phase. Additionally, although we have demonstrated the effectiveness of CaST in various real-world applications, further investigation is required to evaluate its performance in more diverse and complex scenarios.

**Comparison with Existing Works.** *Temporal OoD*: CaseQ [57] and our proposed CaST both segment the temporal environment based on the back-door adjustment to achieve generalization to a new environment. While CaseQ uses shared inference units to associate different types of contexts, CaST stores seen environments in a codebook and approximates unseen environments by the combination of vectors from the codebook. Additionally, CaseQ is not suitable for STG forecasting problems as it is designed for future event prediction and does not consider the spatial context. *Spatial Causal Strength*: The primary challenge in the spatial aspect of this work is how to effectively model the ripple effect of dynamic spatial causal relationships, which is essential for improving node representations for accurate prediction. Although Graph attention [47] and the self-adaptive adjacency matrix proposed by [56] are two alternative methods for learning causal strength, they cannot account for the ripple effect of causation, which involves operating convolution on edges. Specifically, we build a higher-order graph over the edge based on the boundary operator and then use the Hodge-Laplacian operator [16] to filter the edge signal. In Section 5.2, we demonstrate the advantages and effectiveness of incorporating convolution on edge signals to model this effect of causal relationships, and we also provide a case study with visualizations. While there are alternative methods for edge convolution, e.g., the switch of nodes and edges [19, 21], these approaches may face a substantial computational burden, particularly when dealing with non-sparse graphs[16].

**Social Impacts.** By incorporating causal inference, our proposed CaST framework possesses the substantial potential to effect meaningful change across a variety of domains, particularly within the

realm of smart cities. The interpretability offered by CaST promotes a deeper comprehension of complex spatio-temporal systems and their underlying causal dynamics. This integration of causal analysis enables stakeholders to make enlightened decisions, ultimately improving individual and communal quality of life and fostering more resilient, sustainable urban environments. For example, in the context of environmental health, CaST can provide more accurate forecasting of critical air quality indices, like $PM_{2.5}$ concentrations. As air pollution continues to pose significant economic, environmental, and health challenges, the precision of such predictions becomes increasingly paramount. This enhanced accuracy facilitates more effective human health protection measures and informs policy-making, promoting better air quality management and mitigation strategies.

