# OpenReview forum: "Deciphering Spatio-Temporal Graph Forecasting: A Causal Lens and Treatment"
_NeurIPS.cc/2023/Conference — NeurIPS 2023 poster_

### Official Review · Reviewer_ya77 · 2023-06-10

**Soundness:** 2 fair
**Presentation:** 3 good
**Contribution:** 2 fair
**Rating:** 6
**Confidence:** 5

**Summary:**

The paper introduces CaST, a novel framework designed to address challenges in Spatio-Temporal Graph (STG) forecasting. CaST tackles issues related to temporal out-of-distribution and dynamic spatial causation by leveraging a causal lens and employing techniques such as back-door adjustment and front-door adjustment. Experimental results on real-world datasets demonstrate the effectiveness and practicality of CaST, outperforming existing methods while providing good interpretability.

**Strengths:**

Novel design of CaST.
Thorough quantitative and qualitative analysis of the experiment results.



**Weaknesses:**

There are too many design decisions that are made for CaST, making it hard to validate each individual component's contribution to the model performance.
Missing complexity analysis.

**Questions:**

Is the causal structure learned through a data driven approach or based on heuristic reasonings?


**Limitations:**

I am debating if this is a limitation or not. Obviously the proposed method is composed of a lot of sophisticated design components. But maybe just considering causal structure in the time series data can be more compelling and easier to be generalizable.

---

> ### Author Rebuttal · Authors · 2023-08-09
>
> We are grateful for your thorough evaluation of our paper and the feedback you provided. Thank you for recognizing the novelty in the design of our framework and the comprehensiveness of our analysis.
>
> **[Weaknesses]**
>
> **wrt complex design decisions.** Thank you for your feedback. In fact, our main contribution is to approach STG data from a causal perspective and use causal tools to address challenges in STG forecasting, as outlined in our first contribution in Introduction and detailed in Section 3. While the framework may appear relatively intricate, it serves as a vehicle to implement our causal-focused approach, which we believe offers distinctive advantages and presents the core of our contribution. For the validation of these modules in CaST, we have carefully designed a comprehensive ablation study in Section 5 to test the impact of each component on overall performance. For instance, by removing the environment and entity features, we assess the effectiveness of our temporal disentangler in distinguishing these two features.
>
> **wrt complexity analysis.** We apologize for not including this crucial part in our main text. We omit hidden dimensionality in the following analysis for simplicity. The complexity of the spatial module is $ O(M^2)$ induced by the edge convolution operation (i.e., HL Deconfounder) [1], with $M$ denoting the number of nodes; the complexity of the temporal module (i.e., Env Disentangler) is $O(T^2)$, where $T$ represents the historical signal length. Generally, we have $T \ll M$, leading to $O(M^2)$ additional overheads over the original Backbone Encoder. Hope a revision with the above analysis is still considered. Thank you.
>
> [1] Edge representation learning with hypergraphs. NeurIPS 2021.
>
> **[Questions]**
>
> - **Q.** Is the causal structure data-driven or heuristic?
>
>   **A.** Thank you for raising this question. For the CaST framework, the modules’ weights derive from the edge convolution operation, which can be trained through backpropagation. This indicates that our Cast is a data-driven model. As for the SCM, after an extensive literature review, we introduced it specifically for STG data in our scenario, which is also the main contribution of our work.
>
> **[Limitations]**
>
> **wrt sophistication of design components.** Thank you for your thoughtful feedback. As responded in the Weaknesses section, our primary contribution lies in introducing a causal perspective and the novel deep learning framework is used for its implementation. Although CaST comprises multiple components, we believe that each plays a vital role in addressing the challenges we've identified. Beyond the temporal causal effect, it is equally important to model the spatial one considering the highly related spatial correlation in STG data. Thus, to address both the temporal OoD and dynamic spatial causation in STG forecasting, every component is essential.
>
> Again, thank you for your feedback. We have refined our work based on your insights.

---

> > ### Comment · Reviewer_ya77 · 2023-08-17
> >
> > Thanks for sharing your thought. I would keep the current grade.

---

> > > ### Author Response · Authors · 2023-08-19
> > >
> > > Thank you very much for your time and valuable comments!

---

### Official Review · Reviewer_kwLE · 2023-07-03

**Soundness:** 3 good
**Presentation:** 3 good
**Contribution:** 3 good
**Rating:** 6
**Confidence:** 3

**Summary:**

The paper presents a novel framework, CaST, designed to address the challenges of temporal Out-Of-Distribution (OOD) issues and modeling the underlying dynamic spatial causation in Spatio-Temporal Graph (STG) forecasting. The authors construct a Structural Causal Model (SCM) to uncover the causal mechanisms of STG data, inspiring them to employ back-door and front-door adjustments to mitigate the confounding bias induced by temporal environment and spatial context, respectively. Moreover, they utilize the Hodge-Laplacian (HL) operator for edge-level convolution to capture the ripple effect of causation along space and time. Empirical results from experiments on three real-world datasets demonstrate that CaST outperforms existing methods while maintaining interpretability.


**Strengths:**

1. The authors provide an in-depth analysis of the STG forecasting problem, revealing the underlying generation mechanisms of STG data and identifying the key sources of spatio-temporal distribution shift through the lens of causality.
2. The authors offer an interpretation of the specialties of temporal and spatial confounders (i.e., temporal environment and spatial contexts), making the application of backdoor and frontdoor adjustments in CaST targeted and well-motivated.
3. Extensive experiments validate the model's superiority, including comparisons against state-of-the-art STGNNs, ablation studies, hyperparameter sensitivity analysis, and case studies with detailed visualizations of the model's ability to capture spatial causality and identify temporal environments.
4. The figures provide readers with an intuitive understanding of the main problem, key ideas, and compelling experimental results.

**Weaknesses:**

1. The concept of temporal disentanglement is not entirely novel. Although the specific designs of disentanglement headers for environment feature and entity feature may have some novelty, as shown in Figure 4(a), the authors did not adequately clarify this in the main content.

2. For the backdoor adjustment and corresponding temporal disentanglement module: a) The paper lacks a clear explanation regarding the connection between estimating $p(Y|do(X))$ via backdoor adjustment and temporal disentanglement, raising doubts about the effectiveness of the temporal disentanglement module. b) The proposed mutual information regularization cannot ensure that no causal features are leaked into the disentangled environment features $H_e$, possibly leading to the loss of essential causal information.

3. For the frontdoor adjustment and corresponding spatial context filtering module: a) The interpretation of the mediator variable $\hat{H}_i$ is confusing. The authors believe that $\hat{H}_i$ should be "a node representation containing only information propagated based on genuine causation within their spatial context". However, the spatial context $C$ is a confounder, and a mediator containing information from the confounder cannot satisfy the frontdoor criterion due to a backdoor path from the mediator to the label $Y,$ i.e., $\hat{H}_i \leftarrow C \rightarrow Y.$ b) The paper lacks a sufficient explanation regarding the connection between estimating $p(Y|do(X))$ via frontdoor adjustment and spatial context filtering, raising doubts about the effectiveness of the spatial context filtering. c) The reviewer fails to find an explanation for why using the HL operator on edge-graph can learn causation while other GNNs like GAT cannot.

4. The reference [1] addressed the spatio-temporal distribution shift in dynamic graphs. Since the spatio-temporal graph is a special type of dynamic graph, the authors should compare CaST with the approach described in [1].

5. The ablation studies are insufficient. Validating the fact that **w/o Env** and **w/o Ent** can cause performance degradation is unnecessary, as it is a natural result of eliminating information predictive of the label. Additionally, the ablation studies on loss functions corresponding to the two disentanglers are missing.

6. The visualization of dynamic spatial causal relationships in section 5.2 is insufficient since the authors do not check whether other SOTA STGNNs can also capture these relationships.

7. The model's sensitivity to the hyperparameters $\alpha$ and $\beta$, which control the importance of different terms in the final loss function, should be discussed.

[1] Zeyang Zhang, et al. Dynamic Graph Neural Networks Under Spatio-Temporal Distribution Shift. In NIPS 2022.

**Questions:**

1. Why do the authors assume that temporal environment $E$ and spatial context $C$ are independent? For example, the states of neighboring nodes of node i will also be influenced by the change of $E$.
2. How do the proposed modules output statistical estimands that are required to calculate $p(Y|do(X))$ in Eqn. (1) and (2)?
3. Why do the authors focus on learning $p(Y|do(X))$ rather than $p(Y|do(X), E, C)?$ Although $E$ and $C$ are confounders, they can also provide additional information for accurate prediction.
4. In ablation studies, why **w/o Edge**, i.e., replacing 'GCN with causal scores' with GCN, causes such a performance drop in PEMS08? The STGCN also adopts GCN for spatial message passing, but it achieves MAE 18.60 in Table 1.

**Limitations:**

see weakness

---

> ### Author Rebuttal · Authors · 2023-08-09
>
> Thank you for your thorough review and constructive feedback on our paper. We appreciate the time you've taken to provide valuable insights. Please find our responses below.
>
> **[Weaknesses]**
>
> **wrt the novelty of temporal disentanglement.** Thank you for your feedback. We acknowledge that the temporal disentanglement concept has been proposed by previous efforts. And we have highlighted the novelty of implementing the temporal disentangler module in the revision.
>
> **wrt backdoor adjustment and temporal disentanglement.**
>
> - a) We appreciate your attention to this aspect of our work. The relationship between estimating $P(Y|do(X))$ via backdoor adjustment and temporal disentanglement is detailed in Section 4 (please refer to Lines 167-170).
> - b) Regarding your concern about the potential loss of essential causal information, our ablation study in Section 5.2 can address this issue. By testing removing environment or entity features for prediction, our results show that both components offer valuable predictive information, and the entity contains more essential information that aids forecasting. This aligns with our aim for temporal disentangler design: we expect the entity contains more local and vital information like periodicity while the environment contains more global information like the trendy.
>
>
> **wrt frontdoor adjustment and spatial context filtering.**
>
> - a) Thank you for bringing up this insightful query. In our model, $\hat{H}_i$ acts as an estimate for $X^*$, derived from the general causal relationship, while  $X^*$ itself is a surrogate that mimics the deconfouned  $X$. As depicted in Figure 2c, when applying frontdoor adjustment by introducing  $X^*$, it is a cause of $Y$. Consequently, all backdoor paths from  $X^*$  to $Y$ can be blocked (i.e., there are no unmeasured confounders, e.g., $C$,  between  $X^*$ and $Y$). As a result, the relationships  $X^*  \leftarrow C \rightarrow Y $ and $\hat{H}_i\leftarrow C \rightarrow Y $ don't exist.
> - b) The relationship between estimating $P(Y|do(X))$ via frontdoor adjustment and spatial context filtering is detailed in Section 4 (please refer to Lines 171-177).
> -  c) The HL operator was carefully chosen to address the ripple effect of causation, not merely the causation itself.  This choice aligns with the second challenge we aim to tackle in our work, as outlined in the Introduction, Lines 40-46, and Figure 1b. The effectiveness of the HL operator lies in its ability to perform convolution on the graph's edges rather than nodes. The ablation study in Table 2 demonstrates this.
>
>
> **wrt a related model for discussion.** Thank you for bringing the related work DIDA to our attention. It indeed offers significant insights. While both CaST and the DIDA address spatio-temporal distribution shifts, they focus on different tasks: CaST on STG forecasting, and DIDA on link prediction. The distinct nature of these tasks makes direct comparison less meaningful. Nevertheless, we recognize the significance of DIDA and will incorporate this discussion in Section 2.
>
> **wrt hyperparameters sensitivity in the loss function.** Thanks for commenting on this. We've provided additional experiments on the AIR-BJ dataset (see the table below). We find that adjusting the $\alpha$ and $\beta$ weights in the loss function greatly influences the model's performance. We reach a trade-off at $\alpha=0.5$, $\beta=1.5$, and $\alpha=1$, $\beta=1$ to yield the lowest MAE and RMSE, suggesting an optimal trade-off.
>
> |   $\alpha$-$\beta$   |  MAE  |  RMSE |
> |:-------:|:-----:|:-----:|
> | 0.5-0.2 | 23.04 | 35.51 |
> | 0.5-0.5 | 23.06 | 35.41 |
> | 0.5-1.0 | 23.06 | 35.52 |
> | 0.5-1.5 | 22.80 | 34.79 |
> | 1.0-0.2 | 22.93 | 35.23 |
> | 1.0-0.5 | 22.96 | 35.35 |
> | 1.0-1.0 | 22.81 | 34.91 |
> | 1.0-1.5 | 22.95 | 35.33 |
> | 1.5-0.2 | 22.96 | 35.39 |
> | 1.5-0.5 | 22.96 | 35.27 |
> | 1.5-1.0 | 22.96 | 35.33 |
> | 1.5-1.5 | 22.92 | 35.28 |
>
> **wrt visualization of spatial causal relationships.** Thank you for pointing that out. Due to time and space constraints, we'll certainly include such comparisons in the revision.
>
> **[Questions]**
>
> - **Q.** Why do the authors assume the independence of temporal and spatial factors.
>
>   **A.** Thank you for raising this valuable question. Please kindly see our response to Reviewer u4Nz’s Weaknesses Section.
>
> - **Q.** How modules output statistical estimates.
>
>   **A.** Thank you for your question. As outlined in Section 4, our framework indeed implements both Eq.1 and Eq.2. For the former, we employ the Environment Disentangler to isolate environmental features from the input data, and an Environment Codebook to categorize these environments, fulfilling the requirements of Eq.1. As for Eq.2, we use the HL Deconfounder to discern the causal relationships between nodes, assisting in measuring the causal influence of $X$ on $X^*$, thereby meeting the essential criteria of Eq.2.
>
> - **Q.** In ablation studies, why w/o Edge causes such a performance drop in PEMS08?
>
>   **A.** Referencing Line 311, 'w/o Edge' excludes the use of causal scores to guide spatial message passing rather than replacing the process with GCN. This omission naturally leads to a substantial performance drop. While we indeed tested various methods for learning the causal score for spatial message passing (Table 2), where results are only marginally worse than using edge convolution.
>
> Thank you again for your valuable feedback. We have revised our manuscript based on your feedback.

---

> > ### Comment · Reviewer_kwLE · 2023-08-16
> >
> > The rebuttal is acknowledged and I would like to keep the score

---

> > > ### Author Response · Authors · 2023-08-16
> > >
> > > Thank you for taking the time to review our rebuttal and for your thoughtful feedback throughout the review process.

---

### Official Review · Reviewer_u4Nz · 2023-07-06

**Soundness:** 3 good
**Presentation:** 2 fair
**Contribution:** 3 good
**Rating:** 5
**Confidence:** 3

**Summary:**

The paper introduces a model for out-of-distribution prediction in spatio-temporal data. The confounding effects are decoupled in spatial and temporal contexts, and treated with frontdoor and backdoor adjustments, respectively. Empirical evidence shows improved performance.

**Strengths:**

- The tackled problem is extremely relevant.
- To my knowledge, the proposed model consisting of a combination of edge-level filtering, back-door, and front-door adjustment is novel and sound.
- The achieved performance is remarkable.

**Weaknesses:**

- It's unclear to me to what extent the considered assumption of decoupled spatial and temporal environments is reasonable. For instance, the weather (mentioned in line 124) is related to both time and space.
- The experimental setup is not ideal in my opinion.

**Questions:**

### Questions:
- Line 282: Most of the literature considered a different number of steps (see, eg, DCRNN and MTGNN). Although different choices are valid as well, considering the same setting provides a means to assess reproducibility and allows a broader comparison of the methods. Is there a particular reason for not considering these settings?

### Suggestions:
- The top and bottom halves of Fig. 3 referred to in Sec. 4 are not easily identified in the figure. I suggest highlighting them, e.g., with boxes or shaded areas.
- Line 211: This is guaranteed only if the KL divergence is 0. I suggest rephrasing it.
- The causal strength constitutes an important element. I suggest providing a proper definition and discussion about it.
- Other methods in the literature are generally winning over the methods considered in the paper (see e.g. [https://arxiv.org/abs/2005.11650](https://arxiv.org/abs/2005.11650) and the reference therein). I suggest considering some of them, like Graph WaveNet [48] which is also a quite established baseline from 2019.

### Typos
- Line 162: Eq. 8.


### After rebuttal
I have read the author's rebuttal which has addressed all the raised points.

**Limitations:**

yes

---

> ### Author Rebuttal · Authors · 2023-08-09
>
> Thank you for your detailed review of our submission. We appreciate the insights and constructive feedback you have provided. We are grateful for your acknowledgment of the relevance, novelty, and performance of our work. Below, we address your comments in a point-by-point manner.
>
> **[Weaknesses]**
>
> **wrt assumption of decoupled spatial and temporal environments.** Thank you for raising a valid point. Our assumption to treat space and time separately aligns with numerous mainstream ST models, such as GraphWaveNet [1] and STGCN [2]. This treatment optimizes computations and reduces memory consumption, making it practical for real-world applications. It's noteworthy that the performance does not be compromised, as seen with STSGCN [3], where space and time are integrated. Based on these efficiencies and results, we adopted this assumption for our model.
>
> [1] GraphWaveNet for Deep Spatial-Temporal Graph Modeling. IJCAI 2019. (citation: 1,100+)
>
> [2] Spatio-temporal Graph Convolutional Net works: A Deep Learning Framework for Traffic Forecasting. IJCAI 2018. (citation: 2,500+)
>
> [3] Spatial-temporal synchronous graph convolutional networks: A new framework for spatial-temporal network data forecasting. AAAI 2020.
>
>
>
> **wrt experimental setup.**  In the context of air quality prediction, forecasting 12 steps (namely 12 hours) is not practical since citizens usually plan their journey one day ahead. Therefore, we adopted a popular setting from air quality forecasting literature [4], i.e., using the previous 24 steps to predict the next 24 steps. We maintained this setting for PEMS08 for consistency. Furthermore, such longer-term forecasting is more challenging, which poses new hurdles to the prediction model.
>
> [4] Airformer: Predicting nationwide air quality in china with transformers. AAAI 2023.
>
> **[Questions]**
>
> - **Q.** Reason for experiment setting?
>
>   **A.** Please kindly see our response to the Weaknesses Section.
>
>
> **[Suggestion]**
>
> **wrt Figure 3 enhancements.** Thanks for your suggestion. Actually, one of the co-authors also raised this issue before paper submission. We have tried to improve the figure, but we haven't managed to revise it beautifully and clearly yet. We will try our best to improve this figure in the revision. Thank you.
>
>
>
> **wrt KL divergence clarification.** Thank you for pointing out this problem. We've revised the text to: "By minimizing the mutual information, the information overlapping between  $H_e$ and  $H_e$ decreases. When this value approaches zero, each representation is ensured to possess only self-contained information."
>
> **wrt definition on causal strength.** Thank you for your suggestions. We’ve defined ‘causal strength’ as the magnitude of the causal effect between a cause and its outcome. It measures how alterations in one variable directly impact another. A stronger causal strength shows a clear effect, while a weaker one suggests a less obvious relationship. We've included this definition in our revised manuscript.
>
> **wrt considering other methods.** Thank you for your suggestion. Due to the limited time, we've assessed Graph WaveNet [1] on two datasets (i.e., PEMS05 and AIR-BJ) for a comprehensive comparison, as shown in the table below.
>
> | Model         | PEMS08 (MAE)      | PEMS08 (RMSE)    | AIR-BJ (MAE)     | AIR-BJ (RMSE)    |
> |---------------|-----------------|----------------|----------------|----------------|
> | GraphWaveNet  | 16.94 ± 0.43    | 26.70 ± 0.68   | 23.48 ± 0.43   | 36.21 ± 0.69   |
> | CaST (ours)   | **16.44** ± 0.10| **26.61** ± 0.15| **22.90** ± 0.09| **34.84** ± 0.11|
>
>
> **wrt Typo in Line 162.** Thank you for kindly bringing this to our attention. We have rectified this typo in our revision.
>
> Thank you again for your detailed review and insightful feedback. We have meticulously addressed and incorporated your suggestions into our revised manuscript. We believe these are not essentially technical issues. Considering the other three reviewers agree that our paper has good merits such as satisfactory novelty and comprehensive evaluation, we sincerely hope that you could reconsider our score. Thank you so much!

---

> > ### Comment · Reviewer_u4Nz · 2023-08-13
> >
> > I thank the authors for addressing all my concerns.
> >
> > To follow up:
> >
> > > **wrt assumption of decoupled spatial and temporal environments.**
> >
> > Implementation-wise, I agree that it's convenient. However, I am still not fully convinced that this assumption is reasonable for a model meant to be causal -- e.g., due to the weather (line 124).
> >
> > > **wrt experimental setup**  [...] We maintained this setting for PEMS08 for consistency [...]
> >
> > In my opinion, consistency between traffic and air-quality problem setups is not needed. While I see the importance of considering more challenging scenarios, keeping some consistency wrt the literature would have been beneficial for credibility and reproducibility.

---

> > > ### Author Response · Authors · 2023-08-15
> > >
> > > We sincerely thank the reviewer for the follow-up comments.
> > >
> > > **wrt assumption of decoupled spatial and temporal environments.** We appreciate your agreement on the convenience from an implementation standpoint. In the context of our paper regarding this assumption, we acknowledge that certain factors, like weather, can possess both spatial and temporal attributes simultaneously. However, our focus isn't on specific factors per se, but on the impact these factors exert on specific objects — not on the weather as a generalized concept, but its state at a particular location and time. Additionally, there are factors that change only spatially or only temporally, e.g., road networks and working days. Moreover, in the SCM we proposed, the $E$ and $C$ aren't concrete sets with explicitly enumerated factors. Instead, they act as latent, generalized variables. Their role is to encapsulate the broad temporal or spatial effects on $X$ and $Y$, rather than to detail every distinct influencing factor. With this understanding, we believe our assumption about the independence of spatial and temporal effects holds significance. We appreciate your feedback and apologize for any lack of clarity in our paper and previous response.
> > >
> > > **wrt experimental setup.** We sincerely appreciate your feedback. Moving forward, we will strive to strike a balance between practical application scenarios and aligning with the reproducibility standards set by existing literature. Thank you.

---

> > > > ### Comment · Reviewer_u4Nz · 2023-08-17
> > > >
> > > > Thank you again for the elucidations.
> > > > I think that 'borderline accept', where the reasons to accept outweigh the reasons to reject, is still the most appropriate.

---

> > > > > ### Author Response · Authors · 2023-08-17
> > > > >
> > > > > Thank you very much for your time and valuable suggestions and feedback!

---

### Official Review · Reviewer_o65E · 2023-07-06

**Soundness:** 3 good
**Presentation:** 3 good
**Contribution:** 3 good
**Rating:** 6
**Confidence:** 2

**Summary:**

This paper studies the problem of Spatio-Temporal Graph forecasting under the lens of causal treatments. The authors proposed a framework consisting of two major components to mitigate the commonly seen limitations for spatial-temporal GNNs, i.e., 1) the backdoor environment disentanglement block, which models the temporal environmental changes, and 2) the Hodge Laplacian deconfounder capturing the spatial context. The proposed framework is applied in the air and traffic flow datasets to study spatial-temporal interactions.

**Strengths:**

1. [Originality/Significance] The framework is proposed to deal with the limitation of the current STGNN models, i.e., the out of distribution issue and dynamic spatial causation. This paper’s originality comes from a novel combination of causal back-door treatment for temporal components, and a Hodge Laplacian decoupling layer for the spatial contexts.
1. [Quality] The overall experiments presented in this paper for supporting the proposed CaST pipeline are quite nice. It contains different aspects such as ablation study on the core components of the model, which can provide deep insight into the model.
1. [Clarity] The paper is well-written, with detailed descriptions starting from the background, STG data generation causal graph, causal treatment formulation, to each building block in the CaST pipeline.


**Weaknesses:**

1. From Appendix D (L646), it looks like the Hodge Laplacian used here is the down Hodge Laplacian $\partial_1^\top\partial_1$, rather than the full Laplacian (because $\partial_2$ is set to zero here). I would suggest mentioning this in the main text, because this is a big assumption to make. Additionally, there is some existing work using Hodge Laplacian on graphs (e.g., [A]), I would suggest citing this paper for completeness.
1. Related to #1, after applying the edge filter you created, the filtered edge signal will be divergence free [B,C] (due to the low frequency/null space of the down Laplacian being the curl or harmonic flows). This implies that any gradient edge signal (i.e., a flow $x\in\mathbb R^{|E|}$ that can be expressed as $x = \partial_1 y$ for some node function $y \in \mathbb R^{|V|}$) will very likely be filtered out after the convolutional layer, suggesting that the HL deconfounder can learn more information when the flow is divergence free (compared with a curl-free flow). Given that the traffic flow is incompressible (thus divergence free) while air (PM2.5) flow is not, it might also suggest why we see huge drop in performance for the ablation study (Figure 5a) when removing edge signal on the PEM508 (traffic flow) dataset, compared with the AIR_BJ (air flow). I would suggest discussing this assumption/limitation in more detail in the main text.

---
[A] Schaub, Michael T., Austin R. Benson, Paul Horn, Gabor Lippner, and Ali Jadbabaie. “Random Walks on Simplicial Complexes and the Normalized Hodge 1-Laplacian.” SIAM Review 62, no. 2 (2020): 353–91.

[B] Chen, Yu-Chia, Marina Meilă, and Ioannis G. Kevrekidis. “Helmholtzian Eigenmap: Topological Feature Discovery & Edge Flow Learning from Point Cloud Data.” ArXiv:2103.07626 [Stat.ML], March 13, 2021. https://arxiv.org/abs/2103.07626v1.

[C] Schaub, M. T., and S. Segarra. “Flow Smoothing And Denoising: Graph Signal Processing In The Edge-Space.” In 2018 IEEE Global Conference on Signal and Information Processing (GlobalSIP), 735–39, 2018. https://doi.org/10.1109/GlobalSIP.2018.8646701.


**Questions:**

1. Related to the weakness #1, have you tried/considered constructing a $\partial_2$ from e.g., a clique-complex (filling all triangles in the edges), and see if there is performance gain by using the full up and down Hodge Laplacian?
1. In L275, is there any specific reason to regularize only the scaling $\beta$ of $\mathcal L_{mi}$ rather than $\mathcal L_{cod}$? I understand that there is a regularization parameter $\alpha$ in Eq. (6), but how do you make sure the first term of $\mathcal L_{cod}$ is comparable with the log-probability ($\mathcal L_{pre}$)?
1. In L745-750 of Section G, the authors mentioned the edge filter being computationally intensive. If there is space, I would like to see an empirical comparison on the runtime.
1. How should I correctly understand the removal of the Environment codebook in the ablation study (Figure 5a)? Specifically, it seems like the gain for adding Env feature is not huge, does it mean that the environment disentangler is not disentangle the entity from environment well enough (so that some “environmental features” are still presented in the entity features)?


**Limitations:**

The authors have discussed extensively the limitations as well as the social impact in the Appendix, therefore, I do not have any other aspects to add. While I appreciate the detailed discussions in Section G, I believe that it will improve the paper even more if the authors can at least have a brief overview (3-5 sentences) of the limitation/social impact in the main text.

---

> ### Author Rebuttal · Authors · 2023-08-09
>
> We would like to sincerely thank you for the time and effort put into reviewing our submission. Your feedback is invaluable and helps enhance the quality of our paper. We appreciate your acknowledgment of the originality, quality, and clarity of our work. Below, we respond to your comments point-by-point.
>
> **[Weaknesses]**
>
> **wrt the down HL and related work.** We sincerely thank you for the suggestions. We have updated the main text to include this assumption, ensuring clarity for readers. As for the existing work [A], we did cite it for completeness in our revision. Thanks.
>
> **wrt the divergence-free property.** Thank you for your insightful comment. By investigating some related literature in traffic flow theory, we agree that the traffic dataset is incompressible and divergence-free. In this case, message passing is predominantly driven by causal intensity. Conversely, PM$_{2.5}$ in the air quality datasets is compressible [1]. This introduces two challenges: firstly, discerning the real causal score becomes difficult due to potential influences from multiple stations, unlike the explicit relationships between road pairs (i.e., up/downstream influences) within traffic data. Secondly, the vast distances between stations may weaken spatial correlations. We are delighted to add the above discussion to our revision, which considerably strengthens the theory of our model.
>
> [1] Hydrodynamic analysis of compliant foil bearings with compressible air flow[J]. J. Trib., 2004, 126(3): 542-546.
>
> **[Questions]**
>
> - **Q.** Have you tried constructing a $\partial_2$?
> **A.** Thank you for raising the insightful question. We’ve opted against constructing a $\partial_2$ for three primary reasons: (1) our emphasis on primary interactions through causation ripple effects on edges; (2) the lack of explicit meanings for triangles in our task; and (3) reduced computational load. However, considering applications like molecular structure modeling where triangles, such as Benzene Rings, hold significance, a well-defined $\partial_2$ is essential.  In light of your suggestion, we believe it would be valuable to explore the impact of constructing $\partial_2$  on our model's performance in future work.
>
> - **Q.** Why is only scaling $\mathcal{L}_{mi}$?
>
>   **A.** Thank you for your inquiry. We chose not to scale $\mathcal{L}_{cod}$ because its magnitude aligns closely with that of the first term, following the work [2].
>
> - **Q.** The computational intensity and runtime comparison?
>
>   **A.** We appreciate your insightful comment and apologize for any confusion caused due to the lack of clarification in our paper. Indeed, the computational intensity stems from constructing the higher-order (edge) graph from the original dataset, which can be done in the preprocessing stage in one shot. In other words, it won't produce extra computational costs to the training phase. Thus, its impact on the overall training efficiency is negligible. We have followed your advice to clarify this part in our revision.
>
> - **Q.** Does the slight gain from the Env feature suggest issues with the temporal disentangler?
>
>   **A.** Thank you for your insightful question. In designing the temporal disentangler, our intent is to distinctly identify the entity feature, capturing nuances of time series dynamics like periodicity, and the environment feature, aimed at reflecting global trends. As we had foreseen, the removal of the entity causes a significant decrease in performance, while the removal of the environment yields a subtler drop. This aligns precisely with the outcomes presented in our ablation study (see Figure 5a).
>
> [2] Neural discrete representation learning. NeurIPS 2017.
>
> **[Limitations]**
>
> **wrt a brief overview of the limitations/social impact in the main text.** We appreciate your suggestion and have added a brief overview of them correspondingly in our revision.
>
> Once again, thank you for your constructive feedback. We hope our revisions and clarifications address your concerns.

---

> > ### Comment · Reviewer_o65E · 2023-08-17
> >
> > Thank you very much for the detailed response!! I have no further comments/questions. I am confident that this is a good paper for the NeurIPS community, so I will keep the score.

---

> > > ### Author Response · Authors · 2023-08-17
> > >
> > > We sincerely appreciate your constructive feedback and valuable comments, which really contributed to the enhancement of our manuscript. Thank you very much!!

---

### Author Rebuttal · Authors · 2023-08-09

Dear Reviewers,

We would like to express our sincere gratitude to all the reviewers for their thorough evaluation and constructive feedback on our manuscript. Your insights have been invaluable in enhancing the quality and clarity of our work. We have made several revisions accordingly to address the concerns raised:

- **Enhanced Clarifications**: We have enhanced our manuscript with more in-depth explanations where necessary, such as an explanation of ablation studies via a divergence-free perspective and a detailed complexity analysis.

- **Additional Experiments**: Based on the feedback, we have conducted additional experiments to further validate the effectiveness of our proposed model. These include an ablation analysis focusing on hyperparameters within the loss function and a comparison with Graph WaveNet [1].

- **Clearer Definitions**: We have added clear definitions for terms like 'causal strength' to ensure readers have a comprehensive understanding of our method.

- **Expanded Literature Discussion**: We have expanded our discussion on related works, like [2, 3]. This helps in positioning our work in the broader context of the field.

We believe that these revisions have significantly improved our manuscript. We hope that our responses and the changes made address the concerns of the reviewers adequately.

Once again, thank you for your time and effort in reviewing our work. We look forward to your continued feedback.

Best regards,

Authors

**Reference**

[1] GraphWaveNet for Deep Spatial-Temporal Graph Modeling. IJCAI 2019.

[2] Random Walks on Simplicial Complexes and the Normalized Hodge 1-Laplacian. SIAM Review 62, no. 2 (2020): 353–91.

[3] Dynamic Graph Neural Networks Under Spatio-Temporal Distribution Shift. In NIPS 2022.

---

### Decision · Program_Chairs · 2023-09-21

**Decision:**

Accept (poster)

**Comment:**

This paper provides a solid technical contribution to the problem of forecasting on spatio-temporal graphs. Many of the concerns raised by the reviewers were addressed through discussions with the authors.